# Universal thermalization dynamics in (1+1)d QFTs

Richard A. Davison[1] and Luca V. Delacrétaz[2,3]

[1]*Department of Mathematics and Maxwell Institute for Mathematical Sciences, Heriot-Watt University, Edinburgh EH14 4AS, U.K.*

[2]*Kadanoff Center for Theoretical Physics, University of Chicago, Chicago, IL 60637, USA*

[3]*James Franck Institute, University of Chicago, Chicago, IL 60637, USA*

**Abstract**

We identify the universal mechanism behind the thermalization of (1+1)d QFTs at high and low temperatures. Viewing these theories as CFTs perturbed by relevant or irrelevant deformations, we show that conformal perturbation theory in the thermal state breaks down at late times allowing for the emergence of hydrodynamics. This breakdown occurs universally due to the unsuppressed exchange of stress tensors near the lightcone. Furthermore, for theories with central charge $c \to \infty$ we solve for the emergent hydrodynamic theory to all orders in the gradient expansion by arguing that all transport parameters appearing in two-point functions have universal expressions in terms of the scaling dimension $\Delta$ of the perturbation. The radius of convergence of the hydrodynamic dispersion relations provides an early time cutoff for hydrodynamics, which agrees with the time scale at which conformal perturbation theory breaks down.

# 1 Introduction and summary

Interacting systems thermalize, leading to the emergence of hydrodynamics at late times. While the structure of hydrodynamics is universal, *how* a system thermalizes, how long it takes, and the details of the hydrodynamics that emerges, are not. Under certain conditions, these features of quantum field theories (QFTs) may be studied using tools like kinetic theory (at weak coupling) or holography (for a large number of degrees of freedom). In this paper, we will show that low dimensionality offers a complementary way to gain the theoretical control needed to answer these questions.

Thermal correlators in (1+1)d conformal field theory (CFT) are entirely fixed by symmetry, forbidding the emergence of dissipative hydrodynamics. This suggests that (1+1)d QFTs at high and low temperatures thermalize very slowly. In these limits, QFTs can be described as CFTs deformed by a relevant or irrelevant operator $\mathcal{O}$ of dimension $\Delta$:

$$S = S_{\text{CFT}} + \sqrt{c}\lambda \int d^2x\, \mathcal{O}. \tag{1.1}$$

The coupling in units of temperature $\bar{\lambda} \equiv \lambda T^{\Delta-2} \ll 1$ then provides a dimensionless control parameter for conformal perturbation theory (CPT). The anticipated slow thermalization suggests that the real time thermalization dynamics may be analytically tractable in this limit. This is analogous to how kinetic theory captures the slow thermalization of QFTs at weak coupling, but here without any restriction on the nature of the underlying CFT: we expect slow thermalization even when it is strongly coupled.

In this paper, we identify the universal mechanism for the thermalization of (1+1)d QFTs at high and low temperatures. For thermalization to occur the effects of the perturbation in (1.1) must become large. This is because stress tensor correlators have support only near the light front in a CFT, and only near the sound-front in hydrodynamics, and at late times these fronts are far apart. In other words thermalization requires that CPT breaks down at late enough times in the thermal state, even when $\bar{\lambda} \ll 1$. We demonstrate that the exchange of stress tensors near the lightcone causes exactly such a CPT breakdown at times $t \gtrsim \tau_{\text{eq}}$, where

$$\tau_{\text{eq}} \sim \frac{\beta}{\bar{\lambda}^2}, \tag{1.2}$$

and $\beta \equiv 1/T$. Physically, the deformation (1.1) couples the left and right-moving sectors of the CFT, allowing for chiral operators to slow down and form a hydrodynamic sound-front. This process is illustrated in Fig. 1. Interestingly, the timescale (1.2) corresponds parametrically to the fastest thermalization timescale allowed by causality in a (1+1)d QFT [1].

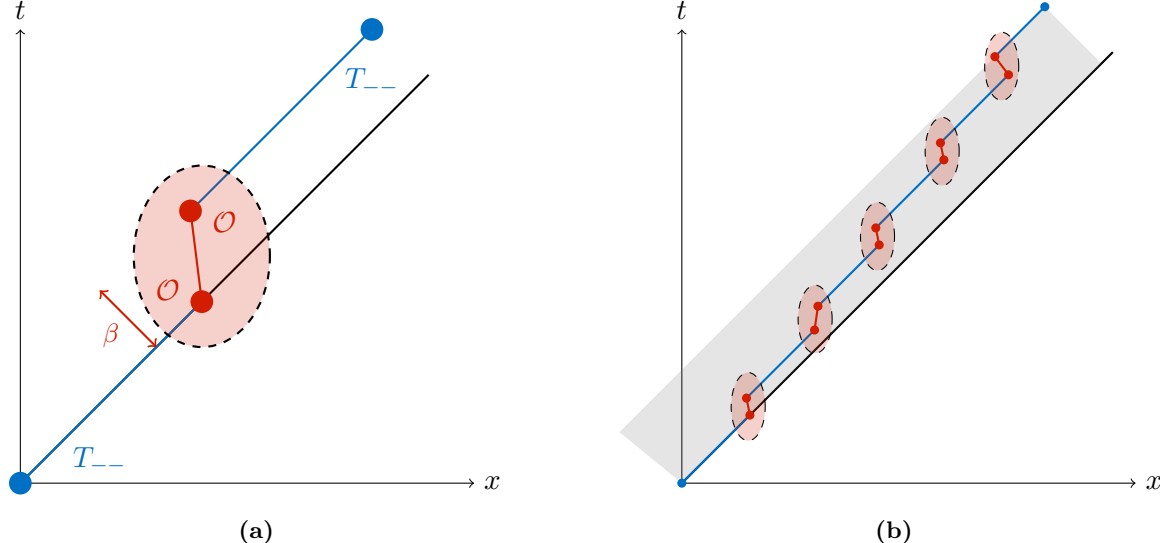

**Figure 1:** In (1+1)d CFT, the thermal two-point function of the stress tensor $T_{\mu\nu}$ is concentrated along the light-front. **(a)** Its first $O(\lambda^2)$ CPT correction can be viewed as two $T_{--}$ propagators convoluted with a $\mathcal{O}$ propagator, cf. Eq. (3.9). This allows the two-point function to become appreciable within a width $\beta$ of the light-front. **(b)** General structure of leading CPT corrections, which allow for the emergence of hydrodynamics at late times $t \gg 1/(\bar{\lambda}^2 T)$. Pairs of $\mathcal{O}$'s (red) fuse into $T_{--}$ and other Virasoro descendants of the identity which propagate along null lines (blue).

The breakdown of CPT should coincide with the emergence of hydrodynamics at the local equilibration (or thermalization) time $\tau_{\text{eq}}$. In the second half of our paper we specialize to cases where the CFT has a large central charge $c$, giving additional theoretical control. In these cases we argue that it is possible to calculate *all* hydrodynamic transport parameters that appear in the two-point functions i.e. to calculate the coefficients of all of the corresponding higher gradient terms in the late-time hydrodynamic effective field theory. The expressions for these are universal: they depend only on the values of $\Delta$ and $T$. From this we reconstruct the hydrodynamic stress tensor two-point functions and the dispersion relations of the hydrodynamic sound waves $\omega_{\pm}(k)$ to all orders in $k$.

Specifically, in momentum space the hydrodynamic retarded Green's function of the trace of the stress tensor is

$$G_R^{\text{trace}}(\omega, k) = (\omega^2 - k^2)\frac{(\varepsilon + P)(1 - c_s^2 - i\omega\Omega(\omega, k^2)) - \frac{c}{12\pi}(\omega^2 - k^2)\kappa(\omega, k^2)}{\omega^2 - c_s^2 k^2 - i\omega k^2 \Omega(\omega, k^2)}, \qquad (1.3)$$

where $\varepsilon$, $P$ and $c_s$ are the energy density, pressure and speed of sound, and $\Omega(\omega, k^2)$ and $\kappa(\omega, k^2)$ are analytic functions around $\omega, k = 0$, whose Taylor coefficients define the transport parameters of the theory. We argue that – to leading order in $\bar{\lambda}$ – each of these parameters

is fixed by the thermal retarded Green's function of the scalar $\mathcal{O}$ in the CFT

$$
\begin{aligned}
((2\pi T)^2 + k^2)(1 - c_s^2 - i\omega\Omega(\omega, k^2)) &- (\omega^2 - k^2)\left(\kappa(\omega, k^2) - 1\right) \\
&= 12\pi\lambda^2(2 - \Delta)^2\left(G_{R,\text{CFT}}^{\mathcal{OO}}(\omega, k) - \frac{\Delta}{(2 - \Delta)}G_{R,\text{CFT}}^{\mathcal{OO}}(0, 0)\right),
\end{aligned}
\tag{1.4}
$$

where the right hand side should be considered as a series in small $\omega$ and $k$ which can be easily calculated from the explicit expression for $G_{R,\text{CFT}}^{\mathcal{OO}}(\omega, k)$ given in Eq. (4.19) below.

This complete knowledge of the hydrodynamic theory allows us to determine when, and why, it breaks down. The hydrodynamic dispersion relations $\omega_\pm(k)$ are analytic functions with a universal radius of convergence $k_{\text{max}} = \Delta\pi T$. At this wavenumber the hydrodynamic poles of momentum space Green's functions collide with poles corresponding to thermal CFT excitations of $\mathcal{O}$. In other words, hydrodynamics breaks down at short scales as it does not account for these excitations. Upon translating back from momentum space to real space, this leads to an early-time cutoff on hydrodynamics that agrees with the time scale (1.2) at which CPT breaks down.

Taken together, our results provide a universal description of thermalization of (1+1)d QFTs at high and low temperatures, from the perspective of both the early-time and late-time effective theories. After reviewing some useful results in Sec. 2, the CPT perspective on thermalization is described in Sec. 3 and the hydrodynamic perspective in Sec. 4. In Sec. 5 we close with some discussion, including some remarks on the prospects of deriving hydrodynamics from CPT.

## 2 Thermodynamics and Ward identities

We study dynamics at finite temperature of $(1 + 1)$-dimensional quantum field theories that are close to conformal fixed points. To do this, we consider the action

$$
S = S_{\text{CFT}} + \sqrt{c}\lambda\int d^2x\,\mathcal{O},
\tag{2.1}
$$

where $S_{\text{CFT}}$ is the action of a conformal field theory with central charge $c$, and $\mathcal{O}$ is a scalar primary operator of this CFT with scaling dimension $\Delta$. We normalize $\mathcal{O}$ such that its vacuum two-point function in the CFT is $|x - x'|^{-2\Delta}$.

We are interested in the limit where the effects of the deformation to the CFT are expected to be small. When $S_{\text{CFT}}$ is a UV fixed point, we take $0 < \Delta < 2$ and consider the high temperature limit $\lambda \ll T^{2-\Delta}$. For field theories that arise by deforming a CFT with multiple relevant scalar operators, the most dominant corrections to CFT physics at higher

temperatures will be due to the least relevant deforming operator. By choosing $\Delta$ to be the dimension of this operator, the simple action (2.1) captures these dominant effects.

The situation is more subtle when $S_{\text{CFT}}$ is an IR fixed point. In this case we consider the low temperature limit $\lambda \ll T^{2-\Delta}$ where the effects of deformations by irrelevant operators with $2 < \Delta$ will be small. RG flow will typically generate deformations by all such operators and provided the least irrelevant has $2 < \Delta < 3$, the simple action (2.1) captures the dominant corrections to CFT physics at low temperatures. The reason for the upper bound on $\Delta$ is that the effects of the $T\bar{T}$ deformation dominate over those of a scalar primary with $\Delta \geq 3$. This is because the enhanced effects of the $T\bar{T}$ operator, which is a descendant of the identity and so has a non-zero expectation value in the thermal state [1]. We treat these special cases, namely the low-temperature dynamics of QFTs that flow to IR CFTs with no operator of dimension $2 < \Delta < 3$, in Sec. 3.3.

## 2.1 Thermodynamics

Thermal expectation values of a CFT on the infinite line can be obtained by mapping Euclidean expectation values from the plane to the cylinder. The corrections due to the dimensionful coupling $\lambda$ can be calculated in conformal perturbation theory and we will parameterize these by the dimensionless coupling

$$\bar{\lambda} \equiv \lambda T^{\Delta-2}, \tag{2.2}$$

which will be small in both limits described above.

The energy density $\varepsilon$ and pressure $P$ of the thermal state are [1]

$$\varepsilon = \frac{\pi c}{6} T^2 \left( 1 + \frac{2\Delta - 3}{1 - \Delta} \alpha_\Delta \bar{\lambda}^2 + \cdots \right), \qquad P = \frac{\pi c}{6} T^2 \left( 1 + \frac{1}{1 - \Delta} \alpha_\Delta \bar{\lambda}^2 + \cdots \right), \tag{2.3}$$

where

$$\alpha_\Delta = 3(2\pi)^{2(\Delta-1)} \frac{\Gamma(2-\Delta) \Gamma(\frac{\Delta}{2})^2}{\Gamma(\Delta) \Gamma(1-\frac{\Delta}{2})^2}, \tag{2.4}$$

and the entropy density is $s = (\varepsilon + P)/T$. The factor of $\sqrt{c}$ in the action (2.1) was chosen so that a coupling $\bar{\lambda} \sim 1$ has a qualitatively important effect on the equation of state.

Defining the speed of sound $c_s$ by $c_s^2 = \frac{d\varepsilon}{dP}$ gives

$$1 - c_s = (2 - \Delta)\alpha_\Delta \bar{\lambda}^2 + \cdots. \tag{2.5}$$

Causality requires that this quantity is non-negative and it is straightforward to verify that this is the case provided $0 \leq \Delta < 3$.

## 2.2 Ward identities

To understand thermalization we will study the two-point functions of the stress tensor $T^{\mu\nu}$ and the primary operator $\mathcal{O}$ that deforms the action. These are strongly constrained by Ward identities. In fact, in (1+1) dimensions there is only one independent two-point function of this set of operators.

To obtain the relations between two-point functions, we first promote the constant coupling $\lambda$ in the action to a spacetime dependent coupling $J(x)$ and the flat metric to $g_{\mu\nu}(x)$. The expectation values in Euclidean signature are then given by

$$\sqrt{g}\langle T^{\mu\nu}\rangle(x) = 2\frac{\delta W}{\delta g_{\mu\nu}(x)}, \qquad\qquad \sqrt{g}\langle\mathcal{O}\rangle(x) = \frac{1}{\sqrt{c}}\frac{\delta W}{\delta J(x)}, \qquad (2.6)$$

where $W[g_{\mu\nu}, J] = -\log Z[g_{\mu\nu}, J]$ is the generator of Euclidean connected correlators (see, e.g., [2,3]). These expectation values obey the Ward identities[1]

$$\nabla_\mu\langle T^{\mu\nu}\rangle = \sqrt{c}\langle\mathcal{O}\rangle\nabla^\nu J, \qquad\qquad \langle T^\mu{}_\mu\rangle = \sqrt{c}(2-\Delta)J\langle\mathcal{O}\rangle + \frac{c}{24\pi}\mathcal{R}. \qquad (2.7)$$

The term involving $\mathcal{R}$ – the Ricci scalar of $g_{\mu\nu}$ – is the Weyl anomaly of the conformal theory. For specific values of $\Delta$, including $\Delta = 1$ and $\Delta = 2$, the trace Ward identity has additional anomalies [4]. We will mostly focus on the generic case given in Eq. (2.7).

We define the connected Euclidean two-point functions by

$$G_E^{\mu\nu\rho\sigma}(x,x') = 2\frac{\delta\left(\sqrt{g}\langle T^{\mu\nu}\rangle(x)\right)}{\delta g_{\rho\sigma}(x')}, \qquad G_E^{\mathcal{OO}}(x,x') = \frac{1}{\sqrt{c}}\frac{\delta\left(\sqrt{g}\mathcal{O}(x)\right)}{\delta J(x')},$$
$$G_E^{\mu\nu\mathcal{O}}(x,x') = \frac{1}{\sqrt{c}}\frac{\delta\left(\sqrt{g}\langle T^{\mu\nu}\rangle(x)\right)}{\delta J(x')}, \qquad G_E^{\mathcal{O}\mu\nu}(x,x') = 2\frac{\delta\left(\sqrt{g}\langle\mathcal{O}\rangle(x)\right)}{\delta g_{\mu\nu}(x')}. \qquad (2.8)$$

These are symmetric under the exchange of the two operators. Taking functional variations of the three Ward identities (2.7) with respect to $J$ and $g_{\mu\nu}$ gives relations between two-point functions. Upon restricting to the flat metric and constant coupling $\lambda$, and then performing a Fourier decomposition

$$G_E(\tau - \tau', x - x') = T\sum_{n\in\mathbb{Z}}\int_{-\infty}^{\infty}\frac{dk}{2\pi}e^{-i\omega_n(\tau-\tau')+ik(x-x')}G_E(\omega_n, k), \qquad \omega_n = 2\pi T n, \quad (2.9)$$

these become algebraic relations, where we are now using $x$ to denote only the spatial coordinate. From solving the relations from the first Ward identity in (2.7) we find that there is only one independent stress tensor two-point function, which we take to be the two-point function of the trace. This is a consequence of the dimensionality. The second

---

[1]We are assuming that the gravitational anomaly vanishes.

Ward identity relates the two-point function of the trace to that of the operator $\mathcal{O}$ that explicitly breaks conformal symmetry. See App. A for more details on these relations.

As thermalization is a real time phenomenon it will be important to work with real time correlators. These can be defined as the analytic continuation of Euclidean time correlators on the thermal cylinder, with different $i\epsilon$ prescriptions leading to different operator orderings [5]. A particularly useful real time correlator is the retarded Green's function, e.g.:

$$G_R^{\mathcal{O}\mathcal{O}}(t,x) \equiv i\theta(t)\langle[\mathcal{O}(t,x),\mathcal{O}]\rangle. \tag{2.10}$$

Its Fourier transform can be shown to analytically continue to the Euclidean Green's function: $G_E(\omega_n, k) = G_R(i\omega_n, k)$. Conversely, $G_R(\omega, k)$ is the only analytic continuation of $G^E(\omega_n, k)$ that is analytic in the upper half plane and does not grow exponentially at large $\omega$, by Carlson's theorem. Following the discussion above, all of the retarded Green's functions of the stress tensor can be expressed in terms of $G_R^{\mathcal{O}\mathcal{O}}$. For example, the two-point function of the trace of the stress tensor is

$$G_R^{\text{trace}}(\omega, k) = -\frac{c}{12\pi}(\omega^2 - k^2) + c\lambda^2(2-\Delta)^2\left(G_R^{\mathcal{O}\mathcal{O}}(\omega, k) - \frac{\Delta}{(2-\Delta)}\frac{\mathcal{O}_{\text{eq}}}{\sqrt{c}\lambda}\right). \tag{2.11}$$

The first term on the right hand side is due to the Weyl anomaly and is the full result in the conformal theory. Indeed, when $\lambda = 0$, the Ward identities of the CFT fix completely all stress tensor two-point functions. The second term is a consequence of conformal symmetry breaking and is exact in $\lambda$. In general $G_R^{\mathcal{O}\mathcal{O}}$ and $\mathcal{O}_{\text{eq}}$ – the expectation value of $\mathcal{O}$ in the thermal state – are functions of $\lambda$.

## 3   Breakdown of conformal perturbation theory

Thermal correlators in (1+1) dimensional CFTs are fixed by symmetry. Breaking of conformal invariance is therefore necessary for a non-trivial hydrodynamic regime to emerge. Even in a QFT obtained by deforming a UV CFT with a relevant operator as in Eq. (2.1), the early time behavior should be described by the CFT in the thermal state, with small corrections accounted for by conformal perturbation theory (CPT). For new physics to emerge at late times, CPT must break down.

CPT was already used to obtain the approximate equation of state at high temperature in (2.3). In these expressions, one expects CPT to fail when $\bar\lambda \gtrsim 1$. However, even when $\bar\lambda \ll 1$, in which case the equation of state is well captured by CPT, we expect CPT to break down for real time correlation functions at late times $t \gtrsim \tau_{\text{eq}}$. We will show in this

Section that the time scale corresponding to the breakdown of CPT for real time thermal correlators is

$$\tau_{\rm eq} \sim \frac{1}{T} \frac{1}{\bar{\lambda}^2}. \tag{3.1}$$

This implies that (1+1) dimensional QFTs at high temperature thermalize as fast as causality allows [1] (this is also true at low temperatures, up to one condition, as will be discussed in Sec. 3.3). For asymptotically free QFTs, this timescale simply corresponds to the mean-free time of particles with cross-section $\sigma \propto \lambda^2$. However, Eq. (3.1) holds for *any* (1+1)d QFT that is UV completed by a CFT. In this more general context, one can think of it as the time scale before which holomorphic factorization, i.e. decoupling of left- and right-moving modes, still effectively holds.

Establishing $\tau_{\rm eq} \sim 1/\lambda^2$ from diagramatics in weakly coupled relativistic theories is difficult, and involves resumming ladder diagrams [6, 7]. The rest of this Section is devoted to the similar task of establishing $\tau_{\rm eq} \sim 1/\lambda^2$ in general (1+1)d QFTs that are close to CFTs, where 'diagramatics' is replaced by real time CPT and the operator product expansion.[2] We will focus for concreteness on the right-moving component of the stress tensor

$$T_{--} , \qquad x^\pm = \frac{1}{\sqrt{2}}(x \pm t) , \tag{3.2}$$

whose thermal two-point function is peaked at the right-moving lightcone $x = t$ in the CFT. As the QFT thermalizes, we expect CPT corrections to become large at late times. We will evaluate CPT corrections to the expected sound-front $x = c_s t$:

$$G_R^{T_{--}T_{--}}(t, x = c_s t) , \tag{3.3}$$

and find that they indeed become large for times larger than $\tau_{\rm eq} = 1/(T\bar{\lambda}^2)$.

## 3.1 Leading CPT correction to stress tensor two-point function

Computing CPT corrections in Eq. (3.3) requires integrating higher-point functions of a CFT over the thermal cylinder. This is challenging even for the leading correction, which requires integrating the CFT four-point function $\langle T_{--}(t, x)T_{--}(0, 0)\mathcal{O}\mathcal{O}\rangle$ twice over the thermal cylinder, once for each $\mathcal{O}$ insertion. However, this leading correction is actually already captured by the dilation Ward identity, Eq. (2.11). We will first look into this leading correction in detail, before going on to study the general structure of CPT corrections. While evaluating the leading correction does not allow to establish the breakdown of CPT, this calculation will already reveal the general pattern that arises at higher orders. It will also

---

[2]See Refs. [8–12] for other uses of CPT in a real time context.

illustrate that CPT corrections can become large at late times, even though the dimensionless coupling is small $\bar{\lambda} \ll 1$.

The two-point function of $T_{--}$ is related to the trace two-point function given in Eq. (2.11) by a diffeomorphism Ward identity (see Eq. (A.4))

$$G_R^{T_{--}T_{--}}(\omega, k) = \frac{(\omega + k)^2}{4(\omega - k)^2} G_R^{\text{trace}}(\omega, k) - (\varepsilon + P)\frac{\omega + k}{\omega - k}. \tag{3.4}$$

Keeping only $O(\lambda^2)$ terms in the trace correlator (2.11) means that $G_R^{\mathcal{OO}}$ should be evaluated in the CFT, and

$$\frac{\mathcal{O}_{\text{eq}}}{\sqrt{c}\lambda} = \frac{1}{c}\partial_\lambda^2 P(T, \lambda)|_{\lambda=0} + \cdots = \frac{\pi}{3}\frac{\alpha_\Delta}{1 - \Delta}T^{2\Delta-2} + \cdots, \tag{3.5}$$

where we used the leading correction to the equation of state (2.3). So the retarded Green's function of the right-moving component of the stress tensor is

$$\frac{1}{c}G_R^{T_{--}T_{--}}(\omega, k) = -\frac{1}{48\pi}\frac{(\omega + k)^3}{\omega - k} - \frac{\pi}{3}\left(1 - \alpha_\Delta\bar{\lambda}^2\right)T^2\frac{\omega + k}{\omega - k} \tag{3.6}$$

$$+ \bar{\lambda}^2 T^2 \frac{(\omega + k)^2}{(\omega - k)^2}\left[\frac{(2 - \Delta)^2}{4T^{2\Delta-2}}G_{R,\text{CFT}}^{\mathcal{OO}}(\omega, k) - \frac{(2 - \Delta)\Delta\alpha_\Delta}{1 - \Delta}\frac{\pi}{12}\right] + O(\bar{\lambda}^3).$$

The first line contains the $O(\lambda^0)$ CFT correlator, whereas the $O(\lambda^2)$ corrections are in the second line and last term of the first line. We will take the inverse Fourier transform of these expressions to study corrections in real time. Note that $G_R$ is analytic in the upper half complex $\omega$ plane, and poles should be resolved by setting $\omega \to \omega + i0^+$. The first line of (3.6) is straightforward and becomes

$$\left[-\frac{1}{6\pi}\delta'''(x - t) + \frac{2\pi}{3}(1 - \alpha_\Delta\bar{\lambda}^2)T^2\delta'(x - t)\right]\theta(t). \tag{3.7}$$

The first term is the vacuum CFT retarded Green's function $\frac{1}{c}G_R^{T_{--}T_{--}}(t, x)$, and the second term is the thermal CFT contribution (together with one simple CPT correction). We are assuming $t > 0$ throughout, and are ignoring contact terms $\propto \delta(t)$ and its derivatives. Both terms above are concentrated on the light front. The only term in (3.6) whose Fourier transform has support in the interior of the lightcone is the term involving a product of $G_R^{\mathcal{OO}}(\omega, k)$ and the factor $f^2(\omega, k) \equiv \frac{(\omega+k)^2}{(\omega-k)^2}$:

$$F(t, x) \equiv \int \frac{d\omega dk}{(2\pi)^2} e^{-i\omega t + ikx} f^2(\omega, k) G_{R,\text{CFT}}^{\mathcal{OO}}(\omega, k). \tag{3.8}$$

There are several ways to evaluate this contribution. The simplest is to use the fact that the Fourier transform of this product is equal to the convolution of Fourier transforms

$$F(t, x) = \int d^2x_1\, G_{R,\text{CFT}}^{\mathcal{OO}}(x^\mu - x_1^\mu)\hat{f}^2(x_1^\mu), \tag{3.9}$$

where $\hat{f}^2$ is the Fourier transform of $f^2(\omega, k) = \frac{(\omega+k)^2}{(\omega-k)^2}$. Here, we will follow a slightly less direct approach to computing (3.8), that will already reflect the general structure of CPT corrections. We will instead view the integrand as a product of *three* factors ($f$, $f$, and $G_R^{\mathcal{O}\mathcal{O}}$), so that its Fourier transform can be written as two convolutions:

$$F(t, x) = \int d^2 x_1 d^2 x_2 \, \hat{f}(x_1^\mu) G_{R,\mathrm{CFT}}^{\mathcal{O}\mathcal{O}}(x_2^\mu - x_1^\mu) \hat{f}(x^\mu - x_2^\mu), \qquad (3.10)$$

with $\hat{f}(t, x) = \delta(t)\delta(x) - 2\partial_x \delta(x - t)\theta(t)$ the inverse Fourier transform of $f(\omega, k) = \frac{\omega+k}{\omega-k+i0^+}$. This has the interpretation of a right-moving stress tensor $T_{--}$ propagating[3] from the origin to $x_1^\mu$, followed by the $\mathcal{O}$ operator propagating from $x_1^\mu$ to $x_2^\mu$, and finally a stress tensor propagating from $x_2^\mu$ to $x^\mu$. This is illustrated in Fig. 1a. While the $\hat{f}$ factors are concentrated along right-moving light-fronts, the scalar Green's function $G_{R,\mathrm{CFT}}^{\mathcal{O}\mathcal{O}}(t, x) = \frac{2\theta(t)\theta(t^2-x^2)\sin(\pi\Delta)}{[(\beta/\pi)^2 \sinh\frac{t-x}{\beta/\pi}\sinh\frac{t+x}{\beta/\pi}]^\Delta}$ allows to move within $\sim \beta$ away from the light-front, as illustrated in red in Fig. 1a.

The largest contribution to (3.10) will come from the long-range $\delta'(x - t)$ piece in both $\hat{f}$ factors. We focus on this contribution here and study the remaining ones, which give subleading corrections at late times, in App. B. After integrating by parts, this contribution to (3.10) is

$$F(t, x) = 4\partial_x^2 \int_0^t dt_1 \int_{t_1}^t dt_2 \, G_{R,\mathrm{CFT}}^{\mathcal{O}\mathcal{O}}(t_2 - t_1, x - t + t_2 - t_1) + \cdots . \qquad (3.11)$$

Changing variables to $t_{\mathrm{av}} = \frac{1}{2}(t_1 + t_2)$ and $t_{21} = t_2 - t_1$, we see that the integrand does not depend on the average location $t_{\mathrm{av}}$ of the pair of operators $\mathcal{O}$. This integral therefore simply produces a factor of $t - t_{21}$, the size of the range of the $t_{\mathrm{av}}$ integral. This factor of $\sim t$ is key to the breakdown of CPT at late times (large $t$). We will see in the next Section that while further CPT corrections are suppressed by $\lambda^2$, they come with additional factors of $t$: the dimensionless number accompanying corrections is $t\bar{\lambda}^2/\beta$, which leads to the breakdown of CPT at times $t \gtrsim \beta/\bar{\lambda}^2$.

Let us finish evaluating (3.11). Inserting the scalar Green's function, one obtains

$$F(t, x) = \frac{8\sin(\pi\Delta)}{(\beta/\pi)^{2\Delta}} \partial_x^2 \frac{1}{\left(\sinh\frac{t-x}{\beta/\pi}\right)^\Delta} \int_{\frac{t-x}{2}}^t dt_{21} \frac{t - t_{21}}{\left(\sinh\frac{x-t+2t_{21}}{\beta/\pi}\right)^\Delta} + \cdots . \qquad (3.12)$$

This integral can be evaluated in terms of hypergeometric functions. However, it is more illuminating to approximate it in the kinematic region of interest, the forward lightcone at late times $t \gg \beta$. One can then replace the upper limit of integration by $t \to \infty$ up to

---

[3]Indeed, notice that the thermal piece of the stress tensor two-point function in the CFT in (3.6) is $\propto \frac{\omega+k}{\omega-k}$.

exponentially small terms $\sim e^{-t/\beta}$. This gives, to leading order in $t \gg \beta$,

$$F(t,x) = \frac{\Gamma\left(\frac{1-\Delta}{2}\right)\Gamma\left(\frac{\Delta}{2}\right)\sin(\pi\Delta)}{\sqrt{\pi}(\beta/\pi)^{2\Delta-1}}(t+x)\partial_x^2 \frac{1}{\left(\sinh\frac{t-x}{\beta/\pi}\right)^\Delta} + \cdots . \tag{3.13}$$

Returning to (3.6), one finds that the final result for the retarded Greens function in the interior of the forward lightcone $t > x$ is:

$$\frac{1}{c}G_R^{T_{--}T_{--}}(t,x) =$$
$$\bar{\lambda}^2 \frac{1}{\beta^2} \frac{(2-\Delta)^2\Gamma\left(\frac{1-\Delta}{2}\right)\Gamma\left(\frac{\Delta}{2}\right)\sin(\pi\Delta)}{4\sqrt{\pi}\pi^{2-2\Delta}}\left[\frac{t+x}{\beta/\pi} - \frac{\pi/2}{\tan\frac{\pi\Delta}{2}}\right]\partial_x^2\frac{1}{\left(\sinh\frac{t-x}{\beta/\pi}\right)^\Delta} \tag{3.14}$$
$$+\; O(e^{-t/\beta}) \;+\; O(\bar{\lambda}^3) \;.$$

We have included subleading terms (the second term in square brackets) and are now precise about the error terms: this expression holds up to exponentially suppressed corrections at late times $t \gg \beta$, and up to higher orders in CPT (see App. B for details).

As anticipated, this leading CPT correction allows $G_R^{T_{--}T_{--}}(t,x)$ to have support away from the strict light-front, at a distance $\beta \gtrsim t - x > 0$ (Fig. 1a). Beyond this qualitative effect, one can also compare more quantitatively this correction to the Wightman function of $T_{--}$, which has support in the interior of the lightcone even in the CFT

$$\langle T_{--}(t,x)T_{--}\rangle_{\text{CFT}} = \frac{c}{2\pi^2}\left(\frac{\pi/\beta}{\sinh\left(\frac{\pi}{\beta}(t-x)\right)}\right)^4 \sim \frac{1}{\beta^4}, \tag{3.15}$$

where in the last step we evaluated at $t - x \sim \beta$. The correction (3.14) in this regime scales as $(\bar{\lambda}^2 t/\beta) \times \frac{1}{\beta^4}$ at late times $t \gg \beta$. It therefore becomes comparable to the leading term when $t$ approaches $\tau_{\text{eq}} = \beta/\bar{\lambda}^2$. This shows that CPT corrections have the potential to become large at late times, even when the coupling is small $\bar{\lambda} \ll 1$.

## 3.2   General scaling of CPT corrections

We will now try to understand the general structure of CPT corrections in the hydrodynamic regime. We are interested in $\bar{\lambda} \ll 1$, where CPT provides a controlled expansion for the equation of state. In particular, the speed of sound $c_s = 1 - O(\bar{\lambda}^2)$ is given by (2.5). In hydrodynamics, one therefore expects a fairly large correlator along the sound-front $x = c_s t$, decaying only polynomially at asymptotically late times (see Eq. (4.2) later). This is clearly not the behavior of the CFT two-point function (3.15), nor of the leading CPT correction to it (3.14): while these decay polynomially on the sound-front for $\beta \ll t \ll \beta/\bar{\lambda}^2$, they decay

exponentially at later times. The emergence of hydrodynamics in (1+1)d QFTs is therefore only visible at higher order in CPT.

We will argue here that there is a proliferation of CPT corrections at times $t \gtrsim \beta/\bar{\lambda}^2$, which allows for the emergence of hydrodynamics. Rather than perform CPT in Euclidean signature and analytically continue to obtain real-time correlators, we will directly study CPT in Lorentzian signature. In Lorentzian two-point functions, the dominant CPT corrections correspond to the deformation $\lambda\mathcal{O}$ integrated over the causal diamond between the two points (the shaded region in Fig. 1b). See App. B.2 for more details on this.

It is difficult to evaluate a general $O(\lambda^n)$ CPT correction to the correlation function (3.3). However, it is fairly simple to identify the dominant channels that will contribute at late times. First, notice that the thermal $\mathcal{O}$ two-point function in the CFT is very 'short-lived': it is exponentially suppressed unless $x \lesssim \beta$ and $t \lesssim \beta$. Therefore we only expect there to be appreciable corrections from configurations where scalars are inserted in spacetime pairs. Furthermore, for such a configuration to produce an appreciable correction to a connected correlator at late times $t \gg \beta$ near the right-moving sound-front, each pair must be located close to this front and fuse into an operator which is long-lived along it. The only such operators (barring an extended current algebra in the CFT) are those in the Virasoro multiplet of the identity, including the stress tensor $T_{--}$ and other chiral descendants. Indeed, the CFT two-point function of $T_{--}$ (3.15) shows that $T_{--}$ is fairly long-lived along the sound-front $x = c_s t = t(1 - O(\bar{\lambda}^2))$: it is only polynomially decaying for $t \lesssim \beta/\bar{\lambda}^2$. To produce a contribution to the correlator (3.3) that is not exponentially small at late times $t \gg \beta/\bar{\lambda}^2$, one can consider a $O(\lambda^{2n})$ CPT correction, with $n \sim t\bar{\lambda}^2/\beta$, where $n$ pairs of $\mathcal{O}$'s fuse into $T_{--}$ (or other Virasoro descendants of the identity) that propagate for a fraction $\sim 1/n$ of the total segment. This is illustrated in Fig. 1b.

How do these higher order CPT corrections scale? Adding a pair of operators $\mathcal{O}$ is suppressed by an additional $\bar{\lambda}^2$; however, while the two operators must be close to each other ($\Delta x, \Delta t \lesssim \beta$), there is freedom in where the pair is positioned along the light front. Integrating that coordinate over the causal diamond produces a factor of $t$. This is exactly what was observed when evaluating the leading correction in the previous Section, below (3.11): while the $t_{21}$ integral was dominated by the region $t_{21} \lesssim \beta$, the integral over the average time $t_{\text{av}}$ of the pair of operators $\mathcal{O}$ produced a factor of $t$. We therefore find that higher order CPT corrections to (3.3) are only suppressed by the dimensionless number

$$\bar{\lambda}^2 t/\beta\,. \tag{3.16}$$

This identifies the time scale $\tau_{\text{eq}} \sim \beta/\bar{\lambda}^2$ at which the CPT corrections that we have described

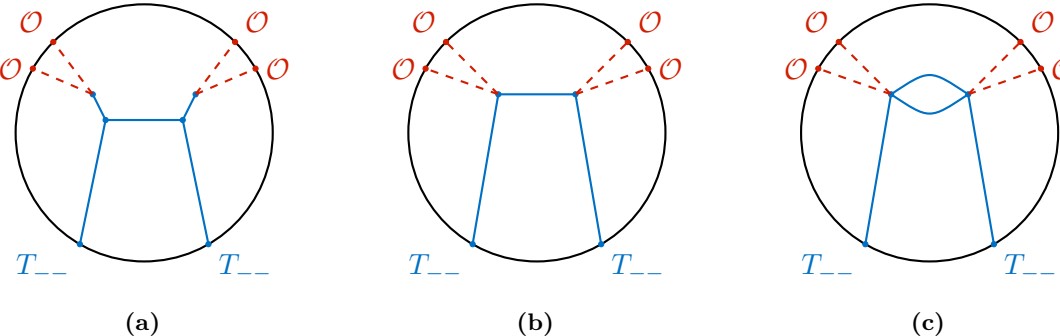

**Figure 2:** Witten diagrams can help identify the large-$c$ scaling of OPE channels, even if the CFT does not have a bulk dual. The figures above show several channels that contribute at $O(\lambda^4)$ to the $T_{--}$ two-point function. Blue lines denote $T_{--}$ (and its global descendants), or "gravitons". **(a)** Both pairs of $\mathcal{O}$'s fuse into $T_{--}$, which then fuse with the external $T_{--}$ into another $T_{--}$. **(b)** Both pairs of $\mathcal{O}$'s instead fuse into $(T_{--})^2$ or other double-twist operators built out of $T_{--}$. **(c)** Pairs of $\mathcal{O}$'s fusing into higher-twist operator such as $(T_{--})^3$ have $1/c$ suppression, as is clear in the diagram which must contain a "graviton loop".

above blow up.

Note that a pair of scalars $\mathcal{O}$ could also fuse into a *left*-mover $T_{++}$ (or descendant), which could give a large correction deeper inside the lightcone. However, in the kinematics considered here ($x \simeq c_s t$), these types of corrections are subleading to the ones identified in Fig. 1b. Indeed, the "left turn" can be placed at any time $t$ and so comes with a factor $\bar\lambda^2 t/\beta$, but the "right turn" must fit in the causal diamond in Fig. 1 and therefore does not have such an enhancement: it comes with a factor $\bar\lambda^2 \times \bar\lambda^2 t/\beta$, where the first term is from inserting the $\mathcal{O}$s and the second is from integration over the causal diamond.

**Large $c$ scaling**

While an explicit computation and resummation of CPT in the hydrodynamic regime seems out of reach, the expansion may be more tractable at large $c$ where conformal blocks simplify [13–15]. From the EFT perspective, the fact that hydrodynamics with $c \to \infty$ (discussed in Sec. 4) is considerably simpler than hydrodynamics with $c < \infty$ makes it seem plausible that one could *derive* the emergence of hydrodynamics from microscopics, for any (1+1)d QFT close to a CFT. We will make here a first step in this direction, by identifying the dominant channels in the general CPT corrections depicted in Fig. 1.

The OPE between two scalars takes the schematic form

$$\mathcal{O}_R \mathcal{O}_R \sim \mathbb{1} + \frac{1}{c} T_{--} + \frac{1}{c^2} (T_{--})^2 + \frac{1}{c^3} (T_{--})^3 + \cdots . \tag{3.17}$$

We are focusing on the holomorphic factors of $\mathcal{O} = \mathcal{O}_L \mathcal{O}_R$, since the left-moving or anti-holomorphic ones will simply fuse into the identity in the leading contributions depicted in Fig. 1b. The stress tensor has a similar OPE, up to an overall factor of $c$, and with one exception:

$$T_{--}T_{--} \sim c \left( \mathbb{1} + \frac{1}{c}T_{--} + \frac{1}{c}(T_{--})^2 + \frac{1}{c^3}(T_{--})^3 + \cdots \right) . \tag{3.18}$$

Notice the enhancement of the $(T_{--})^2$ term, which enters with coefficient 1 in the $T_{--}T_{--}$ OPE. These OPEs allow us to identify which channels give the leading in $c$ contributions to CPT. A convenient way to perform this identification is through Witten diagrams, as shown in Fig. 2. One finds that at leading order in $c$, pairs of $\mathcal{O}$ must fuse into $T_{--}$ or a double-twist operator built out of $T_{--}$, but not any higher-twist operator. The dominant channels at large $c$ therefore seem to consist entirely in products of stress tensor two-point functions.

Let us now compare this to the expected hydrodynamic behavior, obtained in Sec. 4. For $\omega - k = O(\lambda^2)$ and $k \ll T$, we will find (using Eqs. (4.15) and (4.20))

$$G_R^{T_{--}T_{--}}(\omega, k) + sT\frac{\omega + k}{\omega - k} = c\frac{k^2}{\omega - k}\frac{\lambda^2(2 - \Delta)^2 g(k,k) - \frac{1}{12\pi}(\omega^2 - k^2) + \cdots}{\omega - k + \lambda^2 \frac{(2-\Delta)^2}{(\pi/3)T^2}g(k,k) + \cdots} , \tag{3.19}$$

where the $\cdots$ in the numerator and denominator are both $O(\lambda^4)$, and $g(\omega, k) \equiv G_R^{\mathcal{O}\mathcal{O}}(\omega, k) - \frac{\Delta}{2-\Delta}G_R^{\mathcal{O}\mathcal{O}}(0, 0)$. Further expanding this expression in $\lambda^2$ agrees with the leading order CPT correction found in Eq. (3.6). The higher order terms in $\lambda$ appear as a geometric series of scalar propagators near the lightcone $g(k, k)$, times $T_{--}$ propagators $\sim \frac{1}{\omega - k}$. This structure is similar to the one sketched in Fig. 1b, especially at large $c$ where only $T_{--}$ propagators enter. The higher order Witten diagrams (Fig. 2) that form a similar geometric series are the ones that are expected to have a enhancement factor $\propto t$ for every $\mathcal{O}^2$ insertion. Resumming these requires accounting for the exchanged double-twist operators using large-$c$ conformal blocks – we leave this detailed investigation for future work. Another aspect of this discussion that should be improved is that we have used the OPE outside of its strict radius of convergence, in particular when fusing chiral operators along lightrays. It would be interesting to justify this step.

## 3.3   Low temperatures

The equilibrium and out-of-equilibrium dynamics of QFTs at low temperature can also be described by CPT. We will assume that the IR is not gapped, otherwise the thermodynamics is Boltzmann suppressed and thermalization takes an exponentially long time. The IR is

then described by a CFT, with an infinite series of irrelevant corrections

$$S = S_{\text{CFT}_{\text{IR}}} + \sum_i \sqrt{c}\lambda_i \int d^2x\, \mathcal{O}_i + \frac{1}{c}\lambda_{T\bar{T}} \int d^2x\, T\bar{T}\,. \tag{3.20}$$

This can also describe 1+1d lattice systems near a quantum critical point or phase.[4] A notable application in this context is the thermalization of non-linear Luttinger liquids [16].

Among the irrelevant deformations $\mathcal{O}_i$, we have singled out the operator $T\bar{T} \equiv\, :T_{--}T_{++}:$ which can play an important role in the low temperature dynamics of QFTs [1]. The reason is that it is the lightest scalar global primary that is a Virasoro descendant of the identity, so that it can acquire a thermal expectation value $\langle T\bar{T}\rangle = (\pi c/(6\beta^2))^2$. It then already contributes at linear order in CPT $\propto \lambda_{T\bar{T}}$, giving a correction to thermodynamics that is more important than that of operators of dimension $\Delta_i > 3$. The dynamics is then qualitatively different if the dimension of the lightest scalar $\Delta \equiv \min_i \Delta_i$ is greater or lesser than 3. We treat both cases separately below.

**First case:** $2 \leq \Delta \leq 3$

When the lightest irrelevant operator has dimension less than 3, it controls the leading correction to the equation of state and dynamics of the theory at low temperatures $\bar{\lambda} \equiv \lambda T^{\Delta-2} \ll 1$. The analysis so far then essentially goes through without changes. Of course, at very early times correlators are not controlled by the IR CFT, but are sensitive to the UV: these effects can be ignored if $t^2 - x^2 \gg \lambda^{2/(\Delta-2)}$. Away from the lightcone, this requires $t \gg \lambda^{1/(\Delta-2)} = \beta\bar{\lambda}^{1/(\Delta-2)}$; however along the sound-front $x = c_s t = t(1 - O(\bar{\lambda}^2))$, this leads to the stronger condition $t \gg \beta\bar{\lambda}^{\frac{3-\Delta}{\Delta-2}}$. The irrelevant corrections to the CFT are not irrelevant from the perspective of the finite temperature dynamics: they again lead to a breakdown of CPT at later times $t \gtrsim \beta/\bar{\lambda}^2$ as described in Sec. 3.2, allowing for the QFT to thermalize and hydrodynamics to emerge. These regimes are summarized as follows:

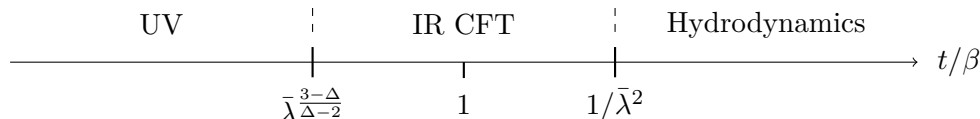

The second and third regimes are captured by our approaches.

---

[4]To fully capture situations where Lorentz invariance is only emergent, one should allow for irrelevant operators in Eq. (3.20) with any Lorentz spin.

**Second case:** $3 < \Delta$

When the least irrelevant correction to the IR CFT has dimension $\Delta > 3$, the leading correction to the equation of state is due to the $T\bar{T}$ operator [1]. The speed of sound at low temperatures is given by [17]

$$c_s = 1 - \frac{\pi}{3}\frac{\lambda_{T\bar{T}}}{\beta^2} + \cdots . \tag{3.21}$$

Subluminality of sound then constrains the coefficient of the $T\bar{T}$ operator to be positive $\lambda_{T\bar{T}} > 0$ (this constraint of course does not apply to lattice UV completions)[5].

The dynamics in this situation is also more subtle. If the irrelevant corrections beyond $T\bar{T}$ are fine-tuned to preserve integrability (the "TTbar" deformation) [17, 19–22], regular hydrodynamics does not emerge. Instead the dynamics is expected to be described by generalized hydrodynamics (GHD) [23–26], the hydrodynamics of systems with a macroscopic number of conserved quantities (see Ref. [27] for a review on GHD). Now even if the higher irrelevant corrections are not fine-tuned to preserve integrability, GHD will describe the dynamics in an intermediate time window before the system ultimately thermalizes and regular hydrodynamics emerges. Let us estimate the time scales where these various regimes describe dynamical correlators such as (3.3) along the sound-front $x = c_s t$, with now $c_s$ given by (3.21). First, as before the expansion in irrelevant operators (3.20) is only controlled at times satisfying $t^2 - x^2 \gg \lambda_{T\bar{T}}$. Along the sound-front, this requires $t \gg \beta$. The correlator is then described by the (thermal) IR CFT (3.15), and has width $\sim \beta$ around the lightcone $x = t$. The TTbar-deformed dynamical correlators have to our knowledge not been computed yet, but we expect them to predict instead a correlator with width $\sim \beta$ around the corrected speed of sound (3.21); this differs significantly from the IR CFT prediction along the sound-front at times $t \gtrsim 1/(\lambda_{T\bar{T}}T^3)$. Finally, integrability is broken by further irrelevant operators which ultimately allows for hydrodynamics to emerge at times $t \gtrsim 1/(\lambda^2 T^3 \times T^{2(\Delta-3)})$ (this is indeed the latest time scale if $\Delta > 3$ – recall that we are assuming $T$ is smaller than all scales entering in the action (3.20)). These regimes are summarized below.

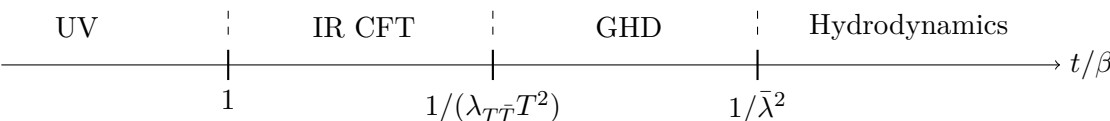

---

[5]See also Ref. [18] for CPT calculations similar to those done in Sec. 3.1, but with $\mathcal{O} = T\bar{T}$.

# 4  Hydrodynamics at large central charge

We have seen how the onset of thermalization is reflected in the breakdown of the CPT that captures the early time dynamics of the system. We are now going to study this phenomenon from the opposite, late time, perspective: by examining how the emergent theory of hydrodynamics breaks down at early times as the effects of thermal CFT excitations become important.

Hydrodynamics in (1+1) dimensions is typically qualitatively different than in higher dimensions as interactions between hydrodynamic modes are more relevant than conventional viscous effects. From now on we will consider the limit of large-$c$ where these interactions are suppressed and the excitations that are relevant at late times are the viscous sound waves familiar from higher dimensions.

Hydrodynamics is an effective theory that – in principle – completely determines the form of the retarded two-point functions of the stress tensor at late times and large distances in terms of transport coefficients. Transport coefficients are the analogue of the Wilson coefficients of an effective field theory. Their values are an input to the theory: they must be determined by some other means and typically depend on details of the specific system.

Specifically, the retarded two-point functions of the stress tensor at late times are peaked around the trajectories of the sound waves $x(t) = \pm c_s t$, with width

$$|x| - c_s t \approx \sqrt{Dt}, \qquad\qquad D = \frac{\zeta}{sT}, \tag{4.1}$$

where the speed of sound $c_s$, the viscosity $\zeta$ and the entropy density $s$ are examples of transport coefficients. Along these trajectories at late times, we will find

$$G_R^{txtx}(t,x) \propto \frac{\exp\left(-\frac{(|x|-c_s t)^2}{2Dt}\right)}{t} \left(1 + \left(1 + \frac{c_s^2 \tau_\Pi}{D}\right) \sqrt{\frac{D}{c_s^2 t}} + \cdots \right), \tag{4.2}$$

where $\tau_\Pi$ – the finite lifetime of pressure perturbations – is another transport coefficient, and we have suppressed unimportant $O(1)$ coefficients for simplicity.

The result (4.2) arises from including only the transport coefficients that are most relevant at late times, but there are infinitely many coefficients providing early-time corrections to this. At a sufficiently early time, we expect that the expansion in (4.2) will become uncontrolled. We identify this timescale as the local equilibration time $\tau_{\rm eq}$, beyond which the system has thermalized and hydrodynamics has emerged. The value of $\tau_{\rm eq}$ can vary widely between different systems since it is determined by the transport coefficients.

It is worth remarking further on our choice to define $\tau_{\rm eq}$ in this way. Firstly, as advocated for in [28], this $\tau_{\rm eq}$ is a property of the hydrodynamic theory itself, and can be defined

independent of any microscopic details. This is in contrast to the $\tau_{\text{eq}}$ computed in Sec. 3, which was defined by the late-time breakdown of a specific UV theory. It is the former that is more universal. In the theories we consider we will see that these two timescales agree, signifying that in these cases the breakdown of CPT coincides with the emergence of hydrodynamics (i.e. there is no intermediate regime governed by a different effective theory – one exception is discussed in Sec. 3.3). See Ref. [1] for an argument that hydrodynamics emerges at $\tau_{\text{eq}} \sim \bar{\lambda}^2/T$ even outside the large-$c$ limit.

Secondly, in defining $\tau_{\text{eq}}$ we are singling out corrections along a specific trajectory – the sound front. One can argue that this is the only sensible definition for $\tau_{\text{eq}}$ (in the absence of additional hydrodynamic modes): the stress tensor two-point function along other rays $x = vt$, $v \neq c_s$ is exponentially suppressed at late times, and highly sensitive to higher gradient terms in hydrodynamics. It is interesting that the timescale $\tau_{\text{eq}}$ thus defined, i.e. the time scale at which the leading corrections in (4.2) along the sound front become large, is not related in a straightforward way to the transport coefficient $\tau_\Pi$.

In this Section we are going to show that there is a surprisingly dramatic simplification near large-$c$ conformal fixed points in (1+1) dimensions. Firstly, we will give a simple classification of all transport coefficients that appear in the stress tensor two-point functions. We will then argue that at small $\bar{\lambda}$ the value of every such transport coefficient is universal – i.e. determined only by the temperature and the values of $c$ and $\Delta$ – and propose a simple generating function from which they can all be easily computed. Armed with this, we will study the breakdown of hydrodynamics at early times by examining momentum space Green's functions. We will establish the local equilibration time $\tau_{\text{eq}} \sim \bar{\lambda}^2/T$, and identify that the breakdown is due to the importance of thermal CFT excitations at sufficiently early times.

## 4.1 Hydrodynamics to all orders

In a system that has thermalized, the excitations that are relevant at late times are those that transport the densities of conserved charges and so are protected from decay by symmetries. In our case, these densities are the energy and momentum densities that obey the local conservation law

$$\nabla_\mu \langle T^{\mu\nu} \rangle = 0, \tag{4.3}$$

where we allow the spacetime metric to be non-trivial for now. Hydrodynamics is the effective theory governing the dynamics of these densities – see [29] for a pedagogical introduction.

The assumption of local equilibration means that the expectation values of all other

operators can be expressed in a derivative expansion in the conserved densities and the spacetime metric. To do this while making Lorentz invariance manifest, it is convenient to reparameterize the stress tensor in terms of auxiliary hydrodynamic variables: a local energy density $\epsilon(x)$ and a local velocity $u^\mu(x)$ where $u_\mu u^\mu = -1$. More precisely, we use the Landau frame condition

$$u_\nu T^{\mu\nu} = -\epsilon u^\mu, \tag{4.4}$$

and then express the stress tensor in terms of the hydrodynamic variables via the constitutive relation

$$T^{\mu\nu} = \epsilon u^\mu u^\nu + P \Delta^{\mu\nu} + \Pi^{\mu\nu}, \qquad\qquad \Delta^{\mu\nu} = g^{\mu\nu} + u^\mu u^\nu. \tag{4.5}$$

The first two terms in $T^{\mu\nu}$ in Eq. (4.5) comprise ideal hydrodynamics, and $P$ – the pressure – is a function of $\epsilon$ that varies between systems and must be input accordingly. The remaining term $\Pi^{\mu\nu}$ is determined order-by-order in an expansion of derivatives of the hydrodynamic variables and the metric. Every term with the appropriate symmetries is included in this expansion, multiplied by its own transport coefficient (a function of the energy density $\epsilon$ that varies between systems).

Once $\Pi^{\mu\nu}$ has been specified, the local conservation equations (4.3) and the constitutive relations (4.5) are a closed set of equations that can be solved for the stress tensor on a given spacetime. In practice, $\Pi^{\mu\nu}$ is typically only calculated to a low order in the derivative expansion as the number of terms proliferates rapidly [30–33]. We are going to show that in (1+1) dimensions, and restricting to small amplitude perturbations around the static equilibrium state, it is possible to compute the constitutive relation to all orders in derivatives. The restriction to small amplitudes will allow us to compute all two-point functions of the stress tensor, but not higher-point functions.

The first simplifications are due to the dimensionality. $\Pi^{\mu\nu}$ is a symmetric tensor satisfying $u_\nu \Pi^{\mu\nu} = 0$. In (1+1) dimensions such a tensor has only a single independent component. Therefore the constitutive relation is specified by a single scalar function

$$\Pi^{\mu\nu} = \Delta^{\mu\nu}\Pi, \tag{4.6}$$

where $\Pi(\nabla^\mu, g_{\mu\nu}, u_\mu, \epsilon, R_{\mu\nu\rho\sigma})$ and $R_{\mu\nu\rho\sigma}$ is the Riemann tensor. In (1+1) dimensions the only independent component of the Riemann tensor is the Ricci scalar $\mathcal{R}$ and so this simplifies to $\Pi(\nabla^\mu, g_{\mu\nu}, u_\mu, \epsilon, \mathcal{R})$.

The second simplifications are achieved by a change of hydrodynamic variable from local energy density $\epsilon(x)$ to $\log(s(x))$, the logarithm of the local entropy density. In these

variables, the conservation equations are

$$D \log(s) = -\nabla_\perp \cdot u + \cdots, \qquad\qquad Du^\mu = -c_s^2 \nabla_\perp^\mu \log(s) + \cdots, \qquad (4.7)$$

where we have decomposed $\nabla^\mu = \nabla_\perp^\mu - u^\mu D$ into the longitudinal and transverse derivatives

$$D \equiv u^\mu \nabla_\mu, \qquad\qquad \nabla_\perp^\mu \equiv \Delta^{\mu\nu} \nabla_\nu, \qquad (4.8)$$

and where $\cdots$ denote higher-derivative corrections to ideal hydrodynamics [31]. The equations (4.7) can be used to eliminate longitudinal derivatives of $\log(s)$ and $u_\mu$ at any order in the derivative expansion [32]. Therefore we only have to consider terms constructed from the transverse derivatives of these hydrodynamic variables.

With these simplifications, the remaining task is to classify all scalars that can be constructed from $(g_{\mu\nu}, u_\mu; \nabla_\perp^\mu, u_\mu, \log(s); \nabla_\mu, \mathcal{R})$. The third simplification comes from considering only small amplitude perturbations around an equilibrium state: a metric of the form

$$g_{\mu\nu}(t, x) = \eta_{\mu\nu} + \delta g_{\mu\nu}(t, x), \qquad (4.9)$$

and hydrodynamic fields of the form

$$\epsilon(t, x) = \varepsilon + \delta\epsilon(t, x), \qquad\qquad u^\mu(t, x) = \delta_t^\mu + \delta u^\mu(t, x), \qquad (4.10)$$

where $\varepsilon$ is the uniform energy density of the thermal state and $sT = \epsilon + P$. The number of allowed scalars is greatly reduced by restricting to only those that are non-zero at linear order in the perturbation amplitude.

We can now classify the allowed terms in $\Pi$. At any order $n \geq 1$ in the derivative expansion, we can construct the allowed scalars by left-multiplying the building blocks $\nabla_{\perp\mu_1} \cdots \nabla_{\perp\mu_n} \log(s)$, $\nabla_{\perp\mu_1} \cdots \nabla_{\perp\mu_n} u_\nu$, and $\nabla_{\mu_1} \cdots \nabla_{\mu_{n-2}} \mathcal{R}$ by appropriate factors of $g^{\mu\nu}$ and $u^\mu$ and contracting the indices. This is because terms with extra factors of the hydrodynamic fields or Ricci scalar inserted between any derivatives will differ from these only by terms that are products of derivatives and so are non-linear in the perturbation amplitude. Furthermore, in each building block the derivatives can be commuted at the expense of introducing only non-linear terms (this is proven in App. C).

We consider first the $\log(s)$ building block. Since $u_\mu \nabla_\perp^\mu = 0$ identically, non-zero scalars can only be constructed by contracting all indices of the transverse derivatives with metric tensors. This is only possible when $n$ is even. And since the transverse derivative operators commute to linear order in perturbation amplitude, there is only one such scalar: $\nabla_\perp^n \log(s)$.

Now we turn to the $u_\nu$ building block. Again, since $u_\mu \nabla_\perp^\mu = 0$ we must contract all indices of the transverse derivatives with metric tensors. Since the transverse derivatives commute to linear order in perturbation amplitude, for odd $n$ the only possible independent scalar is $\nabla_\perp^{n-1}(\nabla_\perp \cdot u)$ and for even $n$ the only one is $u^\nu \nabla_\perp^n u_\nu$. In fact, as $u^\nu \nabla_{\perp\mu} u_\nu = 0$, this latter possibility is non-linear in the perturbation amplitude and so can be discarded.

Finally we turn to the Ricci scalar building block. In general we can contract this with $m$ copies of the metric tensor and $n - 2 - 2m$ copies of the velocity. This produces terms that have the structure of $m$ copies of $\nabla^2$ and $n - 2 - 2m$ copies of $D$ acting on $\mathcal{R}$. Since $\mathcal{R}$ is already linear in the perturbation amplitude, we can replace $\nabla^2$ and $D$ by the partial derivatives $\partial_x^2 - \partial_t^2$ and $\partial_t$ at the cost of only non-linear corrections. As a consequence, the independent scalars are $\nabla_\perp^{2m} D^{n-2-2m} \mathcal{R}$ for all non-negative integer $m \leq (n-2)/2$.

We have now obtained the set of all independent scalars that can appear in $\Pi$. Before proceeding, it is convenient to reorganize the terms that appear at even $n$. Using the hydrodynamic Eq. (4.7), we show in App. C that

$$\nabla_\perp^n \log(s) = -c_s^{-2} \nabla_\perp^{n-2} D \left( \nabla_\perp \cdot u \right) + c_s^{-2} \nabla_\perp^{n-2} \mathcal{R} + \cdots , \tag{4.11}$$

where the $\cdots$ denote higher derivative or non-linear terms. Therefore for even $n$ we can replace $\nabla_\perp^n \log(s)$ by $\nabla_\perp^{n-2} D \left( \nabla_\perp \cdot u \right)$ in our set of independent scalars, since we already include $\nabla_\perp^{n-2} \mathcal{R}$ in this set.

We now write the constitutive relation for $\Pi$ to all orders in the derivative expansion as the sum of all independent scalars outlined above, each multiplied by an independent transport coefficient. This organizes neatly into two independent terms

$$\Pi = (\varepsilon + P)\hat{\Omega}(D, \nabla_\perp^2)\left( \nabla_\perp \cdot u \right) + \frac{c}{24\pi} \hat{\kappa}(D, \nabla_\perp^2) \mathcal{R}, \tag{4.12}$$

where $\hat{\Omega}$ and $\hat{\kappa}$ are the differential operators

$$\hat{\Omega}(D, \nabla_\perp^2) = \Omega_1 + \Omega_2 D + \Omega_3 \nabla_\perp^2 + \Omega_4 \nabla_\perp^2 D + \Omega_5 \nabla_\perp^4 + \Omega_6 \nabla_\perp^4 D + \cdots ,$$
$$\hat{\kappa}(D, \nabla_\perp^2) = \kappa_{2,0} + \kappa_{3,0} D + \kappa_{4,0} D^2 + \kappa_{4,1} \nabla_\perp^2 + \kappa_{5,0} D^3 + \kappa_{5,1} \nabla_\perp^2 D + \cdots . \tag{4.13}$$

$\Omega_n$ and $\kappa_{n,m}$ are transport coefficients, with $n$ labelling the corresponding order of the derivative expansion and $m$ labelling the number of $\nabla_\perp^2$ operators. To linear order in the small amplitude expansion, we can replace the derivatives $D \to \partial_t$ and $\nabla_\perp^2 \to \partial_x^2$ and take each transport coefficient to be a function of the equilibrium temperature. The prefactors of the differential operators in the constitutive relation (4.12) are simply a convenient choice of normalization where $c$ will later be the CFT central charge. Our transport coefficients are

related to the viscosity $\zeta$ and relaxation time $\tau_\Pi$ in (4.2) and [31] by

$$\Omega_1 = -\frac{\zeta}{\varepsilon + P}, \qquad \Omega_2 = \frac{\zeta \tau_\Pi}{\varepsilon + P}. \qquad (4.14)$$

The result (4.12) for $\Pi$ completely specifies the stress tensor and taking derivatives with respect to $\delta g_{\mu\nu}$ gives the stress tensor two-point functions. This is most easily done in momentum space where, for example, the hydrodynamic two-point function of the trace of the stress tensor is

$$G_R^{\text{trace}}(\omega, k) = (\omega^2 - k^2)\frac{(\varepsilon + P)(1 - c_s^2 - i\omega\Omega(\omega, k^2)) - \frac{c}{12\pi}(\omega^2 - k^2)\kappa(\omega, k^2)}{\omega^2 - c_s^2 k^2 - i\omega k^2 \Omega(\omega, k^2)}. \qquad (4.15)$$

where $\Omega(\omega, k^2) = \hat{\Omega}(-i\omega, -k^2)$ and $\kappa(\omega, k^2) = \hat{\kappa}(-i\omega, -k^2)$. As explained in Sec. 2, in (1+1) dimensions all other stress tensor two-point functions can be reconstructed from this using the Ward identity (4.3). The dispersion relations of the hydrodynamic excitations $\omega_{\text{hydro}}(k)$ are given by the poles of Eq. (4.15): these are independent of the $\kappa_{n,m}$ transport coefficients.

## 4.2 Universal generating function for transport coefficients

The hydrodynamic theory described above applies in general in (1+1) dimensions, provided hydrodynamic fluctuations can be neglected. We are now going to specialize to systems with approximate conformal symmetry, i.e. those described by the action (2.1) with $\bar{\lambda} \ll 1$.

Firstly, when the theory is exactly conformal, $P(\epsilon) = \epsilon$ and a comparison of the constitutive relation (4.5) with the trace Ward identity (2.7) gives the exact result $\Pi = (c/24\pi)\mathcal{R}$. We can therefore think of a CFT stress tensor as formally governed by a theory of hydrodynamics with $c_s^2 = 1$, $\kappa_{2,0} = 1$ and all other transport coefficients vanishing. This is formal in the sense that the Green's function is in this case entirely fixed by symmetries, which are one of the inputs of hydrodynamics. The other hydrodynamic input, the assumption that all operators at late time can be expressed in terms of the two conserved densities (or $\epsilon$ and $u^\mu$), is vacuous for (1+1)d CFTs.

When the conformal symmetry is weakly broken $\bar{\lambda} \ll 1$, we expect a non-trivial hydrodynamic regime to emerge. We will argue below that in this limit, *all* hydrodynamic transport parameters can be *derived*. This will rely on one key assumption, that we describe and motivate below, but were not able to prove.

Our starting point is the momentum space relation (2.11) between the two-point function of the trace and that of the scalar operator that breaks conformal symmetry. In the hydrodynamic limit, the former can be expressed in terms of transport coefficients as

$$G_R^{\text{trace}}(\omega, k) + \frac{c}{12\pi}(\omega^2 - k^2) = \frac{(\varepsilon + P + \frac{c}{12\pi}k^2)(1 - c_s^2 - i\omega\Omega) - \frac{c}{12\pi}(\omega^2 - k^2)(\kappa - 1)}{1 + \frac{k^2}{\omega^2 - k^2}(1 - c_s^2 - i\omega\Omega)}, \qquad (4.16)$$

where, for conciseness, we have suppressed the arguments of $\Omega(\omega, k^2)$ and $\kappa(\omega, k^2)$. Since hydrodynamics is an effective theory, this expression should be understood to be valid in an expansion at small $\omega$, $k$. We subsequently expand this as $\bar{\lambda} \to 0$ and keep only the leading term to obtain

$$G_R^{\text{trace}}(\omega, k) + \frac{c}{12\pi}(\omega^2 - k^2) \to \frac{c}{12\pi}\left((4\pi^2 T^2 + k^2)(1 - c_s^2 - i\omega\Omega) - (\omega^2 - k^2)(\kappa - 1)\right),$$
(4.17)

where on the right hand side we mean the leading small $\bar{\lambda}^2$ contribution to each transport coefficient. More precisely, this latter expansion corresponds to the limit $k\bar{\lambda}^2/(\omega \pm k) \to 0$, i.e. far away from the hydrodynamic poles. In real space this schematically corresponds to the part of the hydrodynamic regime that is far from both lightcones.

We are now going to evaluate this quantity in the opposite order of limits. Expanding $\lambda \to 0$ in Eq. (2.11) yields

$$G_R^{\text{trace}}(\omega, k) + \frac{c}{12\pi}(\omega^2 - k^2) \to c\lambda^2(2 - \Delta)^2\left(G_{R,\text{CFT}}^{\mathcal{OO}}(\omega, k) - \frac{\Delta}{(2 - \Delta)}G_{R,\text{CFT}}^{\mathcal{OO}}(0, 0)\right),$$
(4.18)

at leading order, where we have assumed that the two-point function of $\mathcal{O}$ approaches the thermal CFT result [34]

$$G_{R,\text{CFT}}^{\mathcal{OO}}(\omega, k) = \pi(2\pi T)^{2(\Delta - 1)}\frac{\Gamma(1 - \Delta)}{\Gamma(\Delta)}\frac{\Gamma\left(\frac{\Delta}{2} - \frac{i(\omega + k)}{4\pi T}\right)\Gamma\left(\frac{\Delta}{2} - \frac{i(\omega - k)}{4\pi T}\right)}{\Gamma\left(1 - \frac{\Delta}{2} - \frac{i(\omega + k)}{4\pi T}\right)\Gamma\left(1 - \frac{\Delta}{2} - \frac{i(\omega - k)}{4\pi T}\right)}, \quad (4.19)$$

in this limit. The right hand side of (4.18) is then expanded in the hydrodynamic limit of small $\omega$ and $k$ where it gives a series compatible with (4.17).

The key step is now to assume that the two different orders of limits we have taken commute, giving

$$(4\pi^2 T^2 + k^2)(1 - c_s^2 - i\omega\Omega(\omega, k^2)) - (\omega^2 - k^2)\left(\kappa(\omega, k^2) - 1\right)$$
$$= 12\pi\lambda^2(2 - \Delta)^2\left(G_{R,\text{CFT}}^{\mathcal{OO}}(\omega, k) - \frac{\Delta}{(2 - \Delta)}G_{R,\text{CFT}}^{\mathcal{OO}}(0, 0)\right).$$
(4.20)

The right hand side is the generating function – it should be understood as an expansion in $\omega$ and $k$, and gives expressions for all transport coefficients, i.e. $\Omega_n$ and $\kappa_{n,m}$ identified in Eq. (4.13), to leading order in $\bar{\lambda} \ll 1$.

The assumption that there is a limit in which the hydrodynamic and conformal expressions for the momentum space two-point function agree seems to contradict our conclusion in Sec. 3 that even a small breaking of conformal symmetry is important at late times. We reconcile this apparent contradiction by recalling that the large late time corrections of Sec. 3 are found close to the soundcone, while the regime in which we equate the hydrodynamic

and conformal expressions above corresponds schematically to late times but parametrically far from the lightcone (and soundcone).

In other words, we are assuming that far from the lightcone the scalar two-point function in the weakly deformed theory looks like that of a CFT. We use this to extract expressions for the transport coefficients at small $\bar{\lambda}$. Note that it is important to properly keep track of analytic terms in momentum space (contact terms) to make this identification in (4.20). The transport parameters can then be input to the theory of hydrodynamics to tell us what is happening everywhere in the hydrodynamic regime, including near the lightcone.

Explicitly, the generating function gives the following correction to the conformal value for the speed of sound

$$1 - c_s = (2 - \Delta)\alpha_\Delta \bar{\lambda}^2 + \cdots , \tag{4.21}$$

in agreement with (2.5). Expressing the first order transport coefficient $\Omega_1$ as a viscosity $\zeta$ using (4.14), the generating function gives

$$\zeta = \frac{\pi c}{6} T \frac{(2 - \Delta)^2}{(1 - \Delta)} \alpha_\Delta \cot\left(\frac{\pi \Delta}{2}\right) \bar{\lambda}^2 + \cdots . \tag{4.22}$$

This is never negative provided $\Delta \geq 0$. It is interesting to consider the following ratio of transport parameters, which has a finite limit as $\lambda \to 0$

$$\frac{\zeta}{s} \frac{1}{1 - c_s^2} = \frac{1}{4} \frac{(2 - \Delta)}{(1 - \Delta)} \cot\left(\frac{\pi \Delta}{2}\right) + O(\bar{\lambda}^2) . \tag{4.23}$$

This ratio of transport parameters has been discussed in holographic models in $d > 1$ spatial dimensions (with the replacements $1 - c_s^2 \to \frac{1}{d} - c_s^2$ and $s \to 4\pi\eta$ with $\eta$ the shear viscosity), where it was first conjectured to be bounded below by $\frac{1}{2\pi}$ before violations were found [35–39]. Our results, which do not rely on a holographic construction, show that this ratio *is* bounded from below in the high temperature limit of any (1+1)d QFT that is UV completed by a CFT: in this case, $0 < \Delta \leq 2$ and this ratio is bounded from below by its value at $\Delta = 2$, namely $\frac{1}{2\pi}$. Instead, at low temperatures, Eq. (4.23) implies that this ratio is bounded from *above* by this same value, $\frac{\zeta}{s} \frac{1}{1 - c_s^2} \leq \frac{1}{2\pi}$. Indeed, if $2 \leq \Delta < 3$, then (4.23) can take on any value between $\frac{1}{2\pi}$ and 0. For $\Delta > 3$, we still expect Eq. (4.22) to be valid as we do not expect the TTbar deformation to generate viscous effects. With this assumption, for $\Delta > 3$ the ratio on the left hand side of Eq. (4.23) vanishes as $T^{2(\Delta-3)}$ at low temperatures, because $1 - c_s^2$ is parametrically larger than $\zeta$, see Eq. (3.21). Fig. 3 shows a sketch of the qualitative behavior of the bulk viscosity at high and low temperatures.

The generating function also gives higher order transport coefficients. For example, the

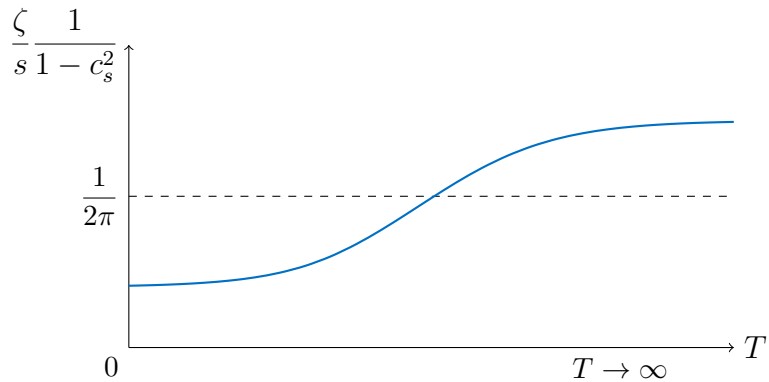

**Figure 3:** Sketch of the possible behavior of the bulk viscosity in a (1+1)d QFT. The leading CPT calculation shows that the ratio $\frac{\zeta}{s}\frac{1}{1-c_s^2}$ is larger than $\frac{1}{2\pi}$ at high temperatures, and smaller than $\frac{1}{2\pi}$ at low temperatures. This ratio is not necessarily monotonic in $T$; see Fig. 6 for examples in holographic theories.

second order transport coefficients are

$$
\begin{aligned}
1 - \kappa_{2,0} &= \frac{(2-\Delta)^2}{2(1-\Delta)}\alpha_\Delta \left( \psi^{(1)}\left(\frac{\Delta}{2}\right) - \frac{\pi^2}{2}\operatorname{cosec}^2\left(\frac{\pi\Delta}{2}\right) + \frac{4(1-\Delta)}{(2-\Delta)} \right) \bar{\lambda}^2 + \cdots, \\
\tau_\Pi &= \frac{\tan\left(\frac{\pi\Delta}{2}\right)}{2\pi^2} \left( \psi^{(1)}\left(\frac{\Delta}{2}\right) - \frac{\pi^2}{2} + \frac{2(1-\Delta)}{(2-\Delta)} \right) \frac{1}{T} + \cdots,
\end{aligned}
\tag{4.24}
$$

where $\psi^{(1)}(z)$ is the polygamma function of order 1, and we have expressed the second order transport coefficient $\Omega_2$ in terms of the timescale $\tau_\Pi$ using (4.14). It is straightforward in principle to continue this procedure to higher orders, but the explicit expressions are not particularly illuminating. One interesting note is that all of these transport coefficients are continuous as $\Delta \to 1$. So although the trace Ward identity is modified for this specific case, it nevertheless seems likely that a more careful calculation would yield the corresponding limit of the answers above.

Our proposal for the simple generating function for transport coefficients (4.20) is really quite remarkable. Even under helpful conditions (e.g. weak coupling or large $N$) the computation of just a single transport coefficient of a QFT is typically difficult and the result sensitive to details of the specific QFT. In contrast, our proposal gives expressions for every transport coefficient that are universal: they depend only on $c$, $T$ and $\Delta$, and are independent of any other details of the theory.

While our proposal is self-consistent, it is obviously important to test it further. We expect the small $k$ and $\lambda$ limits to commute in equilibrium correlation functions due to the finite thermal mass. However, outside of equilibrium it is less clear and so it would

be valuable to compare its predictions to explicit computations in specific QFTs close to a fixed point. In App. D we take a first step in this direction by showing that the viscosity of large-$c$ theories with a holographic dual are indeed given by the expression (4.22). It is clear that our proposal will not be valid outside the large-$c$ limit. At finite $c$, hydrodynamic interactions generate terms in the trace two-point function that are non-analytic in $\omega$ and $k$ and so the expressions on either side of the equality in (4.20) are no longer compatible.[6]

## 4.3  Resummed dispersion relations

At large $c$, the regime of validity of hydrodynamics can be neatly parameterized by the radius of convergence of the dispersion relations of the hydrodynamic excitations [40–43]. We can compute this to leading order in $\bar{\lambda}$ using the generating function.

The excitations of hydrodynamics are sound waves, with dispersion relations given by the poles of Eq. (4.15). It is convenient to represent this as

$$\omega_\pm(k) = \pm k \left(1 + \Gamma_\pm(k)\right), \tag{4.25}$$

such that $\Gamma_\pm(k)$ gives the deviation from the dispersion relations of the thermal CFT. We are interested in $\Gamma_\pm(k)$ at leading order in $\bar{\lambda}^2$, where it is related to the hydrodynamic transport coefficients by

$$\Gamma_\pm(k) = -\frac{1}{2}\left(1 - c_s^2 \mp ik\Omega(\pm k, k^2)\right) + O(\lambda^3). \tag{4.26}$$

This particular combination of transport coefficients can be isolated in the generating function (4.20) by evaluating it at $\omega = \pm k$. Therefore, at leading order in $\bar{\lambda}^2$, the correction to the dispersion relation is controlled by the thermal two-point function of the scalar operator in the CFT

$$\Gamma_\pm(k) = -\frac{6\pi\lambda^2}{(2\pi T)^2 + k^2}(2-\Delta)^2\left(G_{R,\text{CFT}}^{\mathcal{OO}}(\pm k, k) - \frac{\Delta}{(2-\Delta)}G_{R,\text{CFT}}^{\mathcal{OO}}(0,0)\right) + O(\lambda^3). \tag{4.27}$$

The relaxation of the modes is captured by their imaginary part. This is governed by the thermal spectral density of $\mathcal{O}$ in the CFT at $\omega = \pm k$:

$$\text{Im}(\omega_\pm(k)) = \mp\frac{3(2-\Delta)^2}{2\pi}\frac{k}{1+\left(\frac{k}{2\pi T}\right)^2}\text{Im}G_{R,c}^{\mathcal{OO}}(\pm k, k)\left(\frac{\lambda}{T}\right)^2 + O(\lambda^3). \tag{4.28}$$

Equation (4.28) looks temptingly similar to relaxation rates computed in other systems using the memory function formalism [44, 45] and it would be interesting to see if it could be obtained more directly using this approach.

---

[6] For example, the low frequency bulk viscosity of a (1+1)d QFT is $\zeta(\omega)/s \propto \left[\bar{\lambda}^8 T/(c^2\omega)\right]^{1/3}$ [1], showing that the limits $\omega \to 0$ and $\lambda \to 0$ do not commute.

Using the expression (4.19) for the thermal CFT two-point function of a scalar operator gives the explicit dispersion relation

$$\Gamma_{\pm}(k) = -\bar{\lambda}^2 \frac{\Delta(2-\Delta)}{2(1-\Delta)} \frac{\alpha_\Delta}{1 + \left(\frac{k}{2\pi T}\right)^2} \left( \frac{\Gamma\left(2 - \frac{\Delta}{2}\right)\Gamma\left(\frac{\Delta}{2} \mp \frac{ik}{2\pi T}\right)}{\Gamma\left(1 + \frac{\Delta}{2}\right)\Gamma\left(1 - \frac{\Delta}{2} \mp \frac{ik}{2\pi T}\right)} - 1 \right). \qquad (4.29)$$

As always in hydrodynamics, the dispersion relation (4.29) should be understood as a series expansion in $k$. However, the expression written on the right hand side of (4.29) resums this series and so crisply packages information about its convergence. The radius of convergence $k_{\max}$ of the hydrodynamic dispersion relation is determined by the pole of the resummed series that is closest to the origin. For $\Delta \geq 0$ this is always the pole of the gamma function at $k = \mp i\pi T\Delta$, as the apparent poles at $k = \pm i2\pi T$ due to the prefactor are cancelled non-trivially by the terms in brackets. Therefore at small $\bar{\lambda}$ the radii of convergence of the hydrodynamic dispersion relations are $k_{\max} = \Delta\pi T$. For wavenumbers beyond, the hydrodynamic theory is not valid. Just like the transport coefficients, the radius of convergence at small $\bar{\lambda}$ is universal: it depends only on $\Delta$ and $T$.

## 4.4   Breakdown of hydrodynamics

Hydrodynamics is an effective theory whose only excitations are those protected from decay by symmetries. Physically, we expect it to break down at scales where other excitations become important. This is precisely what the radius of convergence is indicating. In this Section we are going to examine in more detail this momentum space picture of the breakdown of hydrodynamics, and how it is consistent with the equilibration timescale $\tau_{\mathrm{eq}} \sim 1/(\bar{\lambda}^2 T)$.

At first glance, the pole in the hydrodynamic dispersion relation (4.29) is problematic as such poles are incompatible with causality [43, 46]. More careful thought reveals that as the wavenumber becomes parametrically close to the pole location $k = \mp i\pi T\Delta + O(\bar{\lambda}^2)$, the $O(\bar{\lambda}^2)$ correction to the dispersion relation (4.29) is parametrically enhanced to $O(\bar{\lambda}^0)$, and thus perturbation theory in $\bar{\lambda}$ is failing. In other words, the pole is an artifact of truncating the dispersion relation at $O(\bar{\lambda}^2)$. Our expectation is that the apparent pole in the dispersion relation at $k = \mp i\pi T\Delta$ is in fact resolved into a branch point at $k = \mp i\pi T\Delta + O(\bar{\lambda}^2)$. Therefore the radius of convergence of the hydrodynamic dispersion relation is

$$k_{\max} = \Delta\pi T + O(\bar{\lambda}^2), \qquad (4.30)$$

where the numerical value of the correction is beyond the scope of our calculation. Branch

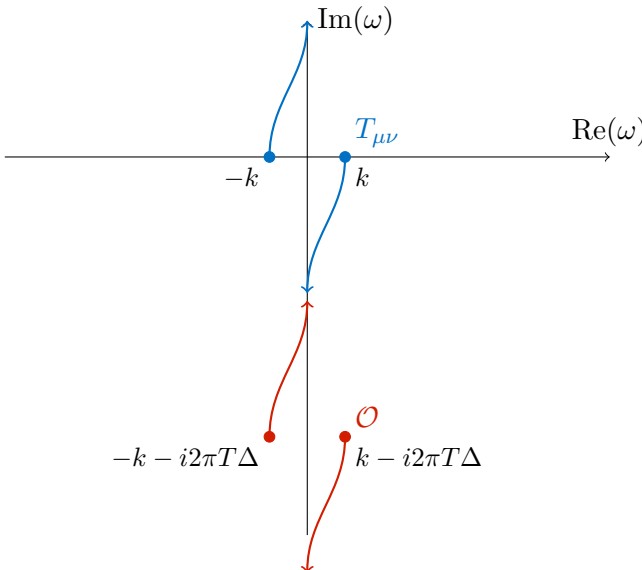

**Figure 4:** Collision in the complex frequency $\omega$ plane of the right-moving hydrodynamic pole (blue) with the left-moving scalar quasi-normal mode (red) as $k$ evolves from a small real number to $-i\pi T\Delta$.

points are compatible with causality and in App. E we provide a more complete analysis of the causality of our dispersion relations.

The existence of a branch point singularity in the dispersion relations is natural upon considering the non-hydrodynamic excitations that become important at short distances. The CFT contains in particular two decoupled types of excitations that are relevant here: the stress tensor two-point function has poles at $\omega = \pm k$ and the scalar two-point function has poles at $\omega = \pm k - i2\pi T(\Delta + 2n)$ with $n = 0, 1, 2, 3, \ldots$. When $\lambda \neq 0$ there will be corrections to these dispersion relations, with the former becoming the hydrodynamic excitations in the appropriate limit. Much more importantly, when $\lambda \neq 0$ the excitations are no longer decoupled: the Ward identity (2.11) ensures that both two-point functions share a common set of poles. Assuming that for $\bar{\lambda} \ll 1$ the corrections to the dispersion relations in the hydrodynamic limit are small, we will argue that it is the coupling of the poles that leads to the breakdown of hydrodynamics.

For concreteness we consider the right-moving sound wave, although an analogous argument applies to the left-moving one. As imaginary $k$ is increased towards $k \to -i\pi T\Delta$, the hydrodynamic pole moves from the origin of the complex $\omega$ plane, directly down the imaginary axis towards $\omega \to -i\pi T\Delta$. Here we are assuming that $\bar{\lambda} \ll 1$ and that we can neglect the corrections to the dispersion relations above in this limit. Under the same

conditions, the right-moving thermal scalar poles move directly down the complex $\omega$ plane from $-i2\pi T(\Delta + 2n)$ to $-i2\pi T(\frac{3\Delta}{2} + 2n)$. In contrast, the left-moving thermal scalar poles move directly *up* in the complex $\omega$ plane from $-i2\pi T(\Delta + 2n)$ to $-i2\pi T(\frac{\Delta}{2} + 2n)$. A sketch of this is shown in Fig. 4. The key point is that the right-moving hydrodynamic pole becomes parametrically close to the $n = 0$, left-moving thermal CFT pole at precisely the wavenumber where we anticipate hydrodynamics breaks down. The natural conclusion is that these poles collide for $k = -i\pi T\Delta + O(\bar{\lambda}^2)$, reflected in a branch point in the hydrodynamic dispersion relation. This is qualitatively similar to the momentum space picture of the breakdown of hydrodynamics in strongly coupled, large $N$ theories in higher dimensions (for example, see [40–42, 47–49]).[7]

The collision of poles has a simple physical interpretation: at short enough scales the thermal scalar excitations of the CFT become important, and hydrodynamics breaks down as it does not account for these. It had to be the case that it was a scalar CFT excitation responsible for the breakdown of hydrodynamics: from Eq. (4.27) the dispersion relation (and so its radius of convergence) is set directly by the scalar CFT thermal two-point function. However, it is non-trivial that it is a collision between opposite-moving modes that leads to a breakdown in hydrodynamics. This further highlights the importance of the coupling between left and right moving modes induced by the breaking of conformal symmetry. The crucial pole collision described above would not occur without such a coupling, and so dissipative hydrodynamics would not emerge. Indeed, the microscopic mechanism identified in 3.2 and Fig. 1b relies on the fact that $\mathcal{O}$ can propagate inside the lightcone.

Finally, armed with our knowledge of the hydrodynamic regime let us return to spacetime to identify the time scale at which this regime emerges. In particular, we will see how a seemingly "Planckian" radius of convergence $k_{\max} \sim T$ leads to a parametrically sub-Planckian thermalization time $\tau_{\mathrm{eq}} \sim 1/(\bar{\lambda}^2 T)$. We are interested in the correlator near the hydroydnamic sound-front, as it is exponentially suppressed at late times elsewhere. Focusing on the right moving front $x \simeq c_s t$, one can Fourier transform (4.15) by first performing the frequency integral and only picking up the right moving sound pole $\omega_+(k) = c_s k - \frac{i}{2} D k^2 + \cdots$.

---

[7]In corresponding weakly coupled theories, kinetic theory calculations (in a relaxation time approximation) indicate that hydrodynamic breakdown is more intricate than simply a pole collision [50–52]. However this intricacy is due to non-hydrodynamic branch points arising from phase space integrals [51], which will not be present in a (1+1)d CFT. In the (1+1) dimensional large-$N$ lattice model studied in [53], a pole collision leads to the breakdown of hydrodynamics even at weak coupling.

The integral over $k$ then yields

$$G_R^{\text{trace}}(t, x = c_s t + \tilde{x}) \simeq A(-i\partial_{\tilde{x}}) \frac{\theta(t)}{\sqrt{2\pi D t}} e^{-\tilde{x}^2/(2Dt)}, \qquad (4.31)$$

where we defined $\tilde{x} \equiv x - c_s t$ and $A$ is a differential operator given by

$$A(k) = \frac{-i(\omega_+^2 - k^2)}{\omega_+ - \omega_-} \left[ sT(1 - c_s^2 - i\omega_+\Omega(\omega_+, k^2)) - \frac{c}{12\pi}(\omega_+^2 - k^2)\kappa(\omega_+, k^2) \right] e^{-i\delta\omega_+ t}.$$

We have suppressed the argument $k$ in the right and left-moving dispersion relations $\omega_\pm(k)$ in this expression, and the last exponent involves $\delta\omega_+ \equiv \omega_+(k) - (c_s k - \frac{i}{2}Dk^2)$. The function $A(k)$ inherits the radius of convergence $k_{\max} = \Delta\pi T$ found in the previous Section. However, each derivative $\partial_{\tilde{x}}$ brings down a factor of $\frac{\tilde{x}}{Dt} \sim 1/\sqrt{Dt}$. Eq. (4.31) therefore is a late-time series expansion in $\frac{1}{k_{\max}\sqrt{Dt}}$. These corrections become small – allowing hydrodynamics to emerge – at the time scale

$$\tau_{\text{eq}} \sim \frac{1}{k_{\max}^2 D} \sim \frac{1}{\bar{\lambda}^2 T}, \qquad (4.32)$$

consistently with what we found in Sec. 3.[8] Inserting the first few terms in the small $k$ expansion of $A(k)$ in (4.31) gives a correlator of the form advertised in Eq. (4.2).

We close this Section with a brief comment on the $\Delta \to 0$ limit of our results. The radius of convergence $k_{\max} = \Delta\pi T$ becomes small in this limit. This arises because the scalar pole in Fig. 4 is closer to the origin – the precocious appearance of "new physics" beyond hydrodynamics lowers the cutoff of the effective hydrodynamic description. However, it is interesting to notice that the equilibration time is not affected by this lowered radius of convergence. Indeed, since (4.22) implies that $D \simeq \frac{12}{\pi^3}\frac{\beta\bar{\lambda}^2}{\Delta^2}$ in this limit, the $\Delta$ dependence drops out of (4.32). This cancellation is also apparent in the microscopic mechanism identified in Sec. 3.2, and arises from the competition of two effects. On one hand, the scalar is longer-lived in the $\Delta \to 0$ limit, $\langle \mathcal{O}(t)\mathcal{O} \rangle \sim e^{-\Delta\pi t/\beta}$, allowing for deeper propagation away from the lightcone in the red regions in Fig. 1 – integrating over the relative coordinate between the two scalars produces a factor of $\sim G_{R,\text{CFT}}^{\mathcal{O}\mathcal{O}}(0,0) \simeq \frac{\beta^2}{\pi\Delta}$. On the other hand, the fusion of scalars into stress tensors $\mathcal{O}\mathcal{O} \sim \frac{\Delta}{c}T + \cdots$ is proportional to $\Delta$, so that the operator $\mathcal{O}$ decouples from the stress tensor in this limit. These two factors cancel, leading to an equilibration time (4.32) that is not singular as $\Delta \to 0$.

---

[8]Alternatively, this time scale can be identified without going through the momentum space Green's functions by expressing the hydrodynamic equations of motion (4.3) in terms of the coordinate $\tilde{x} = x - c_s t$, see Ref. [1].

# 5 Discussion

In summary, we have given a general description of the mechanism and consequences of thermalization in (1+1) dimensional QFTs at high and low temperatures. Thermalization occurs due to the exchange of stress tensors near the lightcone: this leads to the breakdown of conformal perturbation theory at times $t \gtrsim \beta/\bar{\lambda}^2$ and allows for the emergence of dissipative hydrodynamics. At large-$c$, we have argued that the hydrodynamic theory that emerges has universal expressions for the transport coefficients at all orders in the gradient expansion. Analysis of this hydrodynamic theory shows that it breaks down at times $t \lesssim \beta/\bar{\lambda}^2$ where thermal CFT excitations become important. Below we discuss a number of exciting future directions that should be pursued.

**Deriving hydrodynamics:** Rigorous derivations of fluctuating hydrodynamics through the fluctuating Boltzmann equation exist in certain classical models [54–56]. However, hydrodynamics has to our knowledge not been derived for any closed (deterministic) quantum many-body system. The simple structure of the dynamics of (1+1)d QFTs near CFTs, with the additional crutch of the $c \to \infty$ limit, makes it an ideal target for the analytic conformal bootstrap [57, 58] and its large-$c$ implementations [13–15]. It is interesting that the stress tensor plays a prominent role in the dominant channels that we have identified (Sec. 3.2): this does not rely on a sparsity or holographic assumption but follows from its vanishing twist. This suggests that the (double) lightcone limit of the analytic bootstrap, or the lightcone modular bootstrap (see, e.g., [59–61]), may be useful in this regard.

**Tests of proposed hydrodynamics:** As a first concrete step towards this, it is important to establish the validity of our proposed universal generating function for transport coefficients. One way forward would be to derive our hydrodynamic results by alternative methods requiring less assumptions on the details of the effective theory. The obvious candidate is the memory matrix formalism [44, 45], given the structure of some of our results (e.g. Eq. (4.28)). One subtlety is that the memory matrix formalism typically isolates the contributions of long-lived operators to the two-point function, while in our case the dominant contributions come from the trace of the stress tensor which is small, rather than long-lived.

A more direct approach to establishing this is by explicit computation of the transport coefficients in suitable QFTs. In App. D we made a first step in this direction by verifying that the viscosity of holographic theories in the high temperature limit agrees with our proposal. In principle this comparison can be extended beyond just the viscosity to all

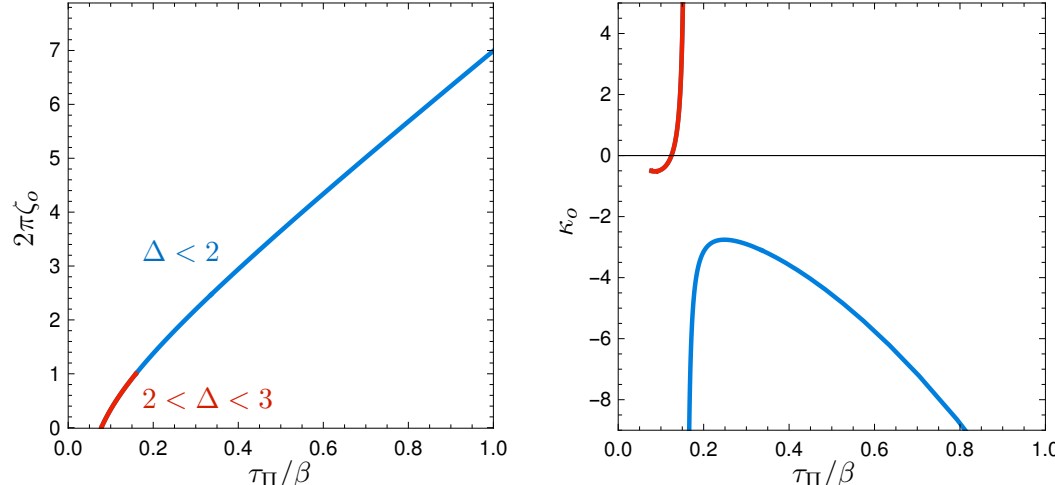

**Figure 5:** Our result (4.23), (4.24) carves out a family of allowed transport parameters, parametrized by $\Delta$ and $T$. The plots above show combinations of parameters where the temperature dependence drops out, as $\bar{\lambda} \to 0$: $\zeta_o \equiv \frac{\zeta}{s} \frac{1}{1-c_s^2}$ (left), and $\kappa_o \equiv \frac{\kappa_{2,0}-1}{1-c_s^2}$ (right), versus $\tau_\Pi/\beta$.

transport coefficients, as well as to the dispersion relations and collisions of the hydrodynamic modes. It can also be extended to IR CFTs, where we expect the competition between the irrelevant $\mathcal{O}$ and TTbar deformations at low temperatures to be realised similarly to analogous phenomena in higher-dimensional theories [62].

Finally, it should also be possible to test our predicted transport parameters in numerics and experiments. We expect recent progress in simulating the out-of-equilibrium dynamics of relativistic (1+1)d QFTs to soon allow access to their hydrodynamic regime [63–67]. Large-$N$ nonlinear sigma models for example could offer an interesting target.[9] Several experimental realizations of 1+1d CFTs exist [68–72] – our approach accounts for corrections away from the CFT, which are inevitable in experiments and should control thermalization and hydrodynamics in these systems.

**Primal hydrodynamic bootstrap:** Progress in UV/IR constraints in QFT (e.g. [73–75]) has renewed the interest in establishing non-perturbative bounds on hydrodynamics transport parameters [1, 28, 46, 76].[10] Ref. [46] in particular found sharp bounds on hydrodynamics in $c \to \infty$ QFTs. Our results in Sec. 4 show that at high and low temperatures, large $c$ hydrodynamics can be solved in (1+1)d: all hydrodynamic transport parameters that appear

---

[9]Note that our conformal perturbation theory approach relies on a finite thermal mass in the 1+1d CFT, and therefore does not apply to a free scalar deformed by $\phi^p$.

[10]A subset of hydrodynamic coefficients already appear in equilibrium thermal effective actions, so that constraining these may prove more tractable [77–79].

in the stress tensor two-point function can be obtained analytically, see Eq. (4.20), and depend smoothly on the dimension $\Delta$ of the perturbation away from the CFT. This can serve the opposite purpose of ruling *in* hydrodynamic theories that have Lorentz invariant UV completions, see Fig. 5. With this in mind it would be worth generalizing our approach to higher-point functions: are the non-linear hydrodynamic transport coefficients similarly fixed by simple CFT data, and if so do CFT constraints place interesting bounds on these?

These results connect more broadly to causality constraints beyond the vacuum, see e.g. [80–83]. It would be interesting in this context to understand the superluminal group velocities observed in App. E that are nonetheless compatible with causality.

**Integrability breaking, GHD, and transport in low dimensions:** The mechanism for thermalization identified in Sec. 3 is somewhat unique to (1+1)d QFTs. It is qualitatively different from weakly coupled QFTs in higher dimension, where transport parameters are non-analytic in the coupling. Viewing (1+1)d CFTs as integrable QFTs [84], the setup we consider has some resemblance with the thermalization of nearly integrable quantum many-body systems. However in those situations, the dynamics is usually described by GHD and is already dissipative before integrability breaking [27]. Moreover, while it is tempting to view (1+1)d QFTs at high and low temperatures as having approximately conserved KdV charges [1], these decouple at large-$c$ so are not generally responsible for slow thermalization.

Transport in (1+1)d is a rich topic [85] and we expect to see further developments in this area. It would be interesting to find classical models with similar thermalization properties as (1+1)d QFTs near CFTs – while hard rods (of negative lengths) [86, 87] or certain celular automata [24, 88] can serve as classical non-relativistic analogues for the TTbar deformation, fixed velocity particles colliding with random time delays may provide a toy model for the thermalization mechanism identified in Fig. 1. In the context of Luttinger liquids, it would also be interesting to distinguish the dynamics between integrability-breaking and preserving deformations [16, 89]. The bulk viscosity is also of interest in nonrelativisitic systems with a large number of degrees of freedom (e.g., [90]).

**Thermalization of (1+1)d CFTs:** The equilibration time marking the onset of dissipative hydrodynamics $\tau_{\mathrm{eq}} \sim \beta/\bar{\lambda}^2$ diverges as one approaches the (1+1)d CFT $\bar{\lambda} \to 0$. Other notions and characterizations of thermalization or chaos may still apply to certain (1+1)d CFTs [91–103]: while emergence of hydrodynamics usually goes hand in hand with other probes of quantum chaos, it could be some of these notions decouple for the case of (1+1)d QFTs close to CFTs. It would be interesting to understand if there is a qualitative difference

between chaotic (1+1)d QFTs obtained by deforming rational or irrational CFTs. This may be possible to investigate in short RG flows that are perturbatively close to minimal models [104–106].

**Higher dimensions:**    Our results fall in line with recent progress in identifying the CFT data that captures thermal physics [77, 78, 107–113]. The case of (1+1) dimensions is special: amongst other reasons, thermal CFT correlators can be computed exactly, the stress tensor sector is formally described by hydrodynamics with almost all transport coefficients vanishing, and thermalization happens parametrically slowly upon deformation of the CFT. On the one hand these features are advantageous as they have allowed us to unravel in some detail the dynamics of thermalization of QFTs. The drawback of course is that many results likely cannot be extended to higher dimensions where conformal symmetry is less restrictive and even an undeformed CFT can thermalize quickly. However we highlight below a couple of aspects that may have higher-dimensional analogues,[11] albeit with less generality.

First, in any dimension there is a Ward identity relating the trace of the stress tensor to the perturbation $\mathcal{O}$. One consequence of this is that many hydrodynamic transport coefficients (e.g. bulk viscosity) are identically zero for an unperturbed CFT [30]. Upon perturbing the CFT, following similar steps as in Sec. 4.2 should give expressions for these transport coefficients in terms of thermal one and two-point functions of $\mathcal{O}$ in the CFT, which may be more readily accessible. This would be a generalization of the approach used in [37] to compute the high temperature bulk viscosity of certain holographic theories.

Second, the breakdown of hydrodynamics that we have found is reminiscent of that in the spatially extended theories with local criticality that arise at low temperatures in large$-N$ Sachdev-Ye-Kitaev chains [115] and black holes with $\mathrm{AdS}_2 \times \mathrm{R}^d$ horizons [116]. In these theories, the radius of convergence is governed by the collision of the hydrodynamic diffusion mode with a thermal excitation of a scalar operator in the critical theory, whose lifetime is determined universally by $T$ and its dimension $\Delta$ [49]. It would be very interesting to see if this relation between the hydrodynamic dispersion relation and the thermal two-point functions of the critical theory could be teased out into expressions for individual transport coefficients like those we have found here.

**Additional global symmetries:**    Finally, we briefly comment on thermalization of (1+1)d QFTs with additional internal global symmetries, such as nonlinear sigma models. In these

---

[11]Analogues in lower dimensions also exist: see, e.g., [114].

situations, the holomorphic factorization in the CFT implies that both the current $j_a^\mu$ and its dual $\tilde{j}_{\mu a} = \epsilon_{\mu\nu} j_a^\nu$ are conserved (see, e.g., [117,118] for related discussions in the context of hydrodynamics). While the former symmetry is exact in the QFT, the latter is only an emergent symmetry at high or low temperatures, and will be broken by the deformation in (1.1). Slow thermalization $\tau_{\text{eq}} \sim 1/(T\bar{\lambda}^2)$ then follows a more familiar pattern of being caused by a long-lived approximately conserved density (the dual density $\tilde{j}^0 = j^x$). The corresponding propagating hydrodynamic mode of the CFT transitions into a diffusive mode at late times, in a way reminiscent of systems with approximately conserved momentum [119]. In this context, we expect the charge diffusivity to be parametrically *large*, $D_c \sim 1/(T\bar{\lambda}^2)$, contrary to the transport parameters identified in Sec. 4.2 that are parametrically small. It would be interesting, in this context, to: (i) identify the dominant CPT corrections, (ii) revisit the commutativity of limits of Sec. 4.2, and finally (iii) study thermal correlators at finite density.

## Acknowledgements

We thank Nathan Benjamin, Abishek Dhar, Sridip Pal, Sav Sethi, Petar Tadić and Matt Walters for helpful discussions and useful comments on the draft. RD is supported by the STFC Ernest Rutherford Grant ST/R004455/1. LD is supported by an NSF award No. PHY2412710.

## A    Ward identities

In this Appendix we spell out how the Ward identities (2.7) fix two-point functions of the stress tensor $T_{\mu\nu}$ and scalar $\mathcal{O}$ in terms of a single structure. Differentiating the first equation, with respect to the metric, then setting $g_{\mu\nu} = \delta_{\mu\nu}$, $J(x) = \lambda = $ constant and finally analytically continuing to real time leads to

$$p_\mu G_R^{\mu\nu\rho\sigma}(p) = -p_\mu \left( \eta^{\rho\nu} \langle T^{\mu\sigma} \rangle + \eta^{\sigma\nu} \langle T^{\mu\rho} \rangle - \eta^{\mu\nu} \langle T^{\rho\sigma} \rangle \right), \tag{A.1}$$

where $p_\mu = (-\omega, k)$. In (1+1) dimensions, this implies that there is only one independent stress tensor two-point function, say that of the trace:

$$G_R^{\mu\nu\rho\sigma}(p) = \frac{\tilde{p}^\mu \tilde{p}^\nu \tilde{p}^\rho \tilde{p}^\sigma}{p^4} G_R^{\text{trace}}(p) + h^{\mu\nu\rho\sigma}(p), \tag{A.2}$$

where $\tilde{p}^\mu = \epsilon^{\mu\nu} p_\nu$, and $h^{\mu\nu\rho\sigma}(p)$ arises due to the contact terms on the right-hand side of (A.1). Using $\langle T^{\mu\nu} \rangle = P\eta^{\mu\nu} + (\varepsilon + P)u^\mu u^\nu$, with $u^\mu = \delta_t^\mu$, one finds that its components in

lightcone coordinates $x^{\pm} = \frac{1}{\sqrt{2}}(x \pm t)$ are given by:

$$h^{---+} = h^{+++-} = -\frac{1}{2}(\varepsilon + P), \qquad h^{++++} = -(\varepsilon + P)\frac{\omega + k}{\omega - k},$$

$$h^{--++} = \varepsilon - P, \qquad h^{----} = -(\varepsilon + P)\frac{\omega - k}{\omega + k}, \tag{A.3}$$

and $h^{+-+-} = 0$ by construction. All other components are fixed by symmetry under $\mu \leftrightarrow \nu$ and $(\mu\nu) \leftrightarrow (\rho\sigma)$. For example this implies that the retarded Green's function of the right-moving component $T_{--}$ is given by[12]

$$\begin{aligned}
G_R^{T_{--}T_{--}}(\omega, k) = G_R^{++++}(\omega, k) &= \frac{(p^+)^4}{(2p^+ p^-)^2} G_R^{\text{trace}}(\omega, k) + h^{++++} \\
&= \frac{(\omega + k)^2}{4(\omega - k)^2} G_R^{\text{trace}}(\omega, k) - (\varepsilon + P)\frac{\omega + k}{\omega - k}.
\end{aligned} \tag{A.4}$$

There are further constraints on the two-point functions from the second Ward identity in (2.7). First, differentiating with respect to the metric and then restricting to the equilibrium state gives the relations

$$\eta_{\rho\sigma} G_R^{\mu\nu\rho\sigma}(p) = \sqrt{c}\lambda(2 - \Delta)G_R^{\mu\nu\mathcal{O}}(p) - 2\langle T^{\mu\nu}\rangle - \frac{c}{12\pi}\tilde{p}^\mu\tilde{p}^\nu. \tag{A.5}$$

Using Eq. (A.2), this becomes an equation for the mixed correlators in terms of the trace two-point function. Second, differentiating with respect to the coupling $J(x)$ and then restricting to the equilibrium state gives

$$\eta_{\mu\nu} G_R^{\mu\nu\mathcal{O}}(p) = \sqrt{c}\lambda(2 - \Delta)G_R^{\mathcal{O}\mathcal{O}}(p) + (2 - \Delta)\mathcal{O}_{\text{eq}}. \tag{A.6}$$

By combining this with the results above, we obtain the key relation (2.11) between the trace two-point function and the $\mathcal{O}$ two-point function.

In principle, there are three further relations that arise by differentiating the first Ward identity in (2.7) with respect to the coupling $J(x)$. However, these are identically satisfied once the conditions above are imposed.

## B Details of CPT calculation

### B.1 Leading correction

The most interesting term in the leading $O(\bar{\lambda}^2)$ CPT correction to the $T_{--}$ two-point function (3.6) is

$$G_R^{T_{--}T_{--}}(\omega, k) \supset \frac{1}{4}c\bar{\lambda}^2(2 - \Delta)^2\beta^{2\Delta - 4} \times f^2(\omega, k)G_R^{\mathcal{O}\mathcal{O}}(\omega, k), \tag{B.1}$$

---

[12]This relates to the normalization used in [120] by $T_{--} = -T_{\text{there}}/\pi$. In our normalization, the leading term in the OPE is $T_{--}(x)T_{--}(0) \sim \frac{c}{8\pi^2}\frac{1}{(x^-)^4} + \cdots$, with $x^- = \frac{1}{\sqrt{2}}(t - x)$.

with $f(\omega, k) = \frac{\omega+k}{\omega-k+i0^+}$. This is the only term in (3.6) that has other non-analyticities than poles at $\omega = k$, which allow its Fourier transform to have support away from the right-moving light-front. We compute its Fourier transform in this Appendix. We will first assume $\Delta < 1$, in which case UV divergences are absent and the Fourier transform of $G_R^{\mathcal{OO}}(x^\mu)$ exists, and discuss $\Delta > 1$ at the end of this Section. Dropping the numerical factor $\frac{1}{4}c\bar{\lambda}^2(2-\Delta)^2\beta^{2\Delta-4}$, we can write its Fourier transform as a double convolution following (3.10):

$$F(t, x) \equiv \int d^2x_1 d^2x_2 \, \hat{f}(x_1^\mu) G_R^{\mathcal{OO}}(x_2^\mu - x_1^\mu) \hat{f}(x^\mu - x_2^\mu) \,, \tag{B.2}$$

with the Fourier transfrom of $f(\omega, k) = \frac{\omega+k}{\omega-k+i0^+}$ given by

$$\hat{f}(t, x) = \delta(t)\delta(x) - 2\partial_x \delta(x-t)\theta(t) \equiv \hat{f}_{\text{ct}}(t, x) + \hat{f}_-(t, x) \,. \tag{B.3}$$

We separated $\hat{f}$ into a contact term $\hat{f}_{\text{ct}}(t, x) = \delta(t)\delta(x)$, and a term $\hat{f}_-$ that has support on the right-moving light-front $x^- = 0$. We can then separate (B.2) into three contributions:

$$F(t, x) = F_{\text{ct, ct}}(t, x) + F_{\text{ct, }-}(t, x) + F_{-, -}(t, x) \,. \tag{B.4}$$

The first is simply $F_{\text{ct, ct}}(t, x) = G_R^{\mathcal{OO}}(t, x)$. We expect the third, $F_{-, -}(t, x)$, to dominate at late times: because it includes two $T_{--}$ "propagators", there is a freedom in where the pair of $\mathcal{O}$'s are placed which leads to an enhancement $\propto t$ for this correction (see Fig. 1a). It was computed to leading order for $t \gg \beta$ in Sec. 3.1; we compute it here up to exponential precision $O(e^{-t/\beta})$, and evaluate $F_{\text{ct, }-}(t, x)$ as well.

To simplify notation, it will be useful to introduce chiral factors $a(t \pm x)$ of the scalar Green's function

$$G_R^{\mathcal{OO}}(t, x) = \frac{2\theta(t-x)\theta(t+x)\sin\pi\Delta}{[(\beta/\pi)^2 \sinh\frac{t-x}{\beta/\pi} \sinh\frac{t+x}{\beta/\pi}]^\Delta} \equiv \frac{\theta(t-x)\theta(t+x)}{a(t-x)a(t+x)} \,. \tag{B.5}$$

Then, the second term in (B.4) is

$$\begin{aligned}
F_{\text{ct, }-}(t, x) &= 2\int d^2x_1 \hat{f}_-(x_1^\mu) G_R^{\mathcal{OO}}(x^\mu - x_1^\mu) \\
&= -4\partial_x \int dt_1 \theta(t_1) G_R^{\mathcal{OO}}(t - t_1, x - t_1) \\
&= -\partial_x \frac{4}{a(t-x)} \int_0^{\frac{t+x}{2}} \frac{dt_1}{a(t+x-2t_1)} \\
&= -\partial_x \frac{2}{a(t-x)} \int_0^{t+x} \frac{ds}{a(s)} \,,
\end{aligned} \tag{B.6}$$

where we changed variables from $t_1$ to $s = t + x - 2t_1$, and dropped an overall factor of $\theta(t-x)$ since we are assuming $t > x$ (otherwise the entire retarded Green's function vanishes

due to causality). We now use the fact that $a(s) \propto \sinh \frac{s}{\beta/\pi} \sim e^{-\pi s/\beta}$ is exponentially small for $s \gg \beta$ to replace the upper limit of integration $t + x \to \infty$, up to exponentially small corrections. Our result is thus

$$F_{\mathrm{ct},-}(t,x) = -\partial_x \frac{2}{a(t-x)} \int_0^\infty \frac{ds}{a(s)} + O(e^{-t/\beta}). \tag{B.7}$$

We now turn to $F_{-,-}(t,x)$:

$$
\begin{aligned}
F_{-,-}(t,x) &= \int d^2x_1 d^2x_1 \hat{f}_-(x_1^\mu) G_R^{\mathcal{OO}}(x_2^\mu - x_1^\mu) \hat{f}_-(x^\mu - x_2^\mu) \\
&= 4\partial_x^2 \int dt_1 dt_2 \theta(t_1)\theta(t-t_2) G_R^{\mathcal{OO}}(t_2 - t_1, x - t + t_2 - t_1) \\
&= \partial_x^2 \frac{4}{a(t-x)} \int_0^t dt_{21} \int_{\frac{1}{2}t_{21}}^{t-\frac{1}{2}t_{21}} dt_{\mathrm{av}} \frac{\theta(x-t+2t_{21})}{a(x-t+2t_{21})} \\
&= \partial_x^2 \frac{4}{a(t-x)} \int_{\frac{t-x}{2}}^t dt_{21} \frac{t-t_{21}}{a(x-t+2t_{21})}.
\end{aligned}
\tag{B.8}
$$

In the third line we changed variables to $t_{21} = t_2 - t_1$ and $t_{\mathrm{av}} = \frac{1}{2}(t_1 + t_2)$; the integral over $t_{\mathrm{av}}$ is trivial and was performed in the last line. Changing variables to $s = x - t + 2t_{21}$ and extending again the upper limit of the integral to $\infty$, one finds up to exponentially small terms:

$$F_{-,-}(t,x) \simeq \partial_x \frac{2}{a(t-x)} \int_0^\infty \frac{ds}{a(s)} + \partial_x^2 \frac{1}{a(t-x)} \left[ (x+t) \int_0^\infty \frac{ds}{a(s)} - \int_0^\infty \frac{s\,ds}{a(s)} \right].$$

The first term exactly cancels the previous result (B.7), so that collecting all contributions and returning to (B.4) we find

$$
\begin{aligned}
F(t,x) &= \partial_x^2 \frac{1}{a(t-x)} \left[ (x+t) \int_0^\infty \frac{ds}{a(s)} - \int_0^\infty \frac{s\,ds}{a(s)} \right] + O(e^{-t/\beta}) \\
&= \frac{\sin(\pi\Delta)}{(\beta/\pi)^{2\Delta-1}} \frac{\Gamma(\frac{1-\Delta}{2})\Gamma(\frac{\Delta}{2})}{\sqrt{\pi}} \left( x + t - \frac{\beta}{2\tan\frac{\pi\Delta}{2}} \right) \partial_x^2 \frac{1}{\left( \sinh\frac{t-x}{\beta/\pi} \right)^\Delta} + O(e^{-t/\beta})
\end{aligned}
\tag{B.9}
$$

Restoring the factor of $\frac{1}{4} c\bar{\lambda}^2(2-\Delta)^2 \beta^{2\Delta-4}$, this gives Eq. (3.14) quoted in the main text.

Let us now comment on UV divergences that arise if $\Delta \geq 1$: in this situation, the Fourier transform of the scalar two-point function (B.5) is UV divergent, a divergence that can be absorbed by adding a (relevant) counterterm to the action $S_{\mathrm{ct}} = \Lambda^{2(\Delta-1)} \int d^2x\, J^2(x)$, where $J(x)$ is the source for $\mathcal{O}$. While the inverse Fourier transform of the CPT correction (B.1) is well-defined, our approach to computing it using a convolution in (B.2) suffers from this UV divergence. The simplest is to analytically continue our final expression (3.14) to $\Delta > 1$. Alternatively, one can directly Fourier transform $\frac{(\omega+k)^2}{(\omega-k+i0^+)^2}$ times the momentum space scalar Green's function (4.19).

## B.2 Real time CPT and causal diamond

Systematizing the study of CPT corrections to higher orders requires integrating $\mathcal{O}$ insertions over the thermal cylinder. This is most naturally done in Euclidean signature:

$$\langle T_{--}(x)T_{--}(0)\rangle_\beta = \sum_n \frac{(-\lambda)^n}{n!} \int\limits_{x_1^\mu,\ldots,x_n^\mu \in S_\beta^1 \times \mathbb{R}} \langle T_{--}(x)T_{--}(0)\mathcal{O}(x_1)\cdots\mathcal{O}(x_n)\rangle_{\beta,\text{CFT}}, \quad \text{(B.10)}$$

followed by continuing the external coordinate $x^\mu$ to real time (all correlators above are connected). However, performing CPT directly in real time allows for better intuition for the channels that are expected to dominate at late times, as discussed in Sec. 3.2. Analytically continuing all coordinates $x_i^\mu$ in the $(n+2)$-point functions appearing in (B.10) produces a fully retarded Green's function involving $n+1$ nested commutators[13] [121, 122]. The integration region is therefore in the past lightcone of $T_{--}(x^\mu)$. This leads to a new complication: while the Euclidean expression Eq. (B.10) is manifestly free of IR divergences,[14] it appears that the channels identified in Fig. 1 can involve $\mathcal{O}$ insertions at arbitrarily early times, which would be IR divergent (since $T_{--}$ CFT correlators are unsuppressed along the lightcone).

In this Section, we consider a formulation of CPT in real time in terms of 'interaction picture' Hamiltonian evolution [123] that makes manifest the absence of IR divergences. This is mostly intended as a sanity check for the mechanism identified in Fig. 1 – we do not necessarily expect that this formulation will make the explicit evaluation of higher CPT corrections more tractable than the Euclidean formulation (B.10). Writing the Hamiltonian as

$$H = H_{\text{CFT}} + \delta H, \qquad \delta H = -c\sqrt{\lambda}\int dx\,\mathcal{O}(t=0,x), \quad \text{(B.11)}$$

interaction picture operators are defined as operators evolving purely in the CFT

$$A_I(t) \equiv e^{iH_{\text{CFT}}t}Ae^{-iH_{\text{CFT}}t}. \quad \text{(B.12)}$$

They are related to regular Heisenberg operators as

$$A(t) = e^{iHt}e^{-iH_{\text{CFT}}t}A_I(t)e^{iH_{\text{CFT}}t}e^{-iHt} \equiv U^\dagger(t)A_I(t)U(t), \quad \text{(B.13)}$$

where the unitary $U(t) \equiv e^{iH_{\text{CFT}}t}e^{-iHt}$ satisfies the equation of motion

$$\partial_t U(t) = -ie^{iH_{\text{CFT}}t}\delta He^{-iHt} = -i\delta H_I(t)U(t), \quad \text{(B.14)}$$

---

[13]In Keldysh notation, this corresponds to $G_{raa\cdots a}$.

[14]This follows from the fact that 2d CFTs have a thermal mass given by the dimension of the lightest operator $m_{\text{CFT}} = \Delta_{\min}/\beta$. Quantizing in the $x$ direction and inserting a complete basis of states in (B.10) shows that it decays as $e^{-m_{\text{CFT}}|x_i-x_j|}$ at large spatial separations $|x_i - x_j| \to \infty$.

which involves the deformation in interaction picture $\delta H_I(t) \equiv e^{iH_{\text{CFT}}t}\delta H e^{-iH_{\text{CFT}}t}$. The solution is

$$U(t) = Te^{-i\int_0^t dt'\,\delta H_I(t)}\,,\tag{B.15}$$

where $T$ denotes time-ordering. One can similarly find a representation for the thermal density matrix as $e^{-\beta H} = e^{-\beta H_0}\times$(expression involving $\delta H$), by defining

$$e^{-\beta H} \equiv e^{-\beta H_0}U_E(\beta)\,,\tag{B.16}$$

One representation for $U_E(\beta) = e^{\beta H_0}e^{-\beta H}$ can be found by solving its imaginary time equation of motion

$$\partial_\beta U_E(\beta) = -\delta H_I^E(\beta)U_E(\beta) \qquad \Rightarrow \qquad U_E(\beta) = T_E e^{-\int_0^\beta d\tau\,\delta H_I^E(\tau)}\,,\tag{B.17}$$

where $T_E$ denotes $\tau$ ordering and

$$\delta H_I^E(\tau) \equiv e^{\tau H_0}\delta H e^{-\tau H_0} = \delta H_I(-i\tau)\,.\tag{B.18}$$

Putting these pieces together, we arrive at the following representation of a real time thermal correlator (ommitting the factor of $Z = \text{Tr}\,e^{-\beta H}$)

$$\langle A(t,x)B\rangle_\beta = \text{Tr}(e^{-\beta H}A(t,x)B)\tag{B.19}$$
$$= \text{Tr}\left(e^{-\beta H_0}\left[T_E e^{-\int_0^\beta d\tau'\,\delta H_I^E(\tau')}\right]\left[\bar{T}e^{i\lambda\int_0^t dt'\,\delta H_I(t')}\right]A_I(t,x)\left[Te^{-i\lambda\int_0^t dt'\,\delta H_I(t')}\right]B_I\right)$$
$$= \left\langle\left[T_E e^{-\lambda\int_0^\beta d\tau'dx'\,\mathcal{O}^E(\tau',x')}\right]\left[\bar{T}e^{i\lambda\int_0^t dt'dx'\,\mathcal{O}(t',x')}\right]A(t,x)\left[Te^{-i\lambda\int_0^t dt'dx'\,\mathcal{O}(t',x')}\right]B\right\rangle_{\beta,\text{CFT}}.$$

The final expression is a correlation function in the CFT. The CPT corrections of interest are those integrated on real times $t' \in [0,t]$. The fact that these integration regions are bounded prohibits IR divergences. Furthermore, expanding both unitaries $U^\dagger(t)$, $U(t)$ above leads to nested commutators with $A(t,x)$, restricting the integration region to the past lightcone of $(t,x)$. Finally, given that $T_{--}$ correlators are peaked along the lightcone, we expect the kinematically dominant region of these integrals to be the causal diamond between $(0,0)$ and $(t,x)$, as depicted in Fig. 1b.

# C  Hydrodynamic constitutive relation

In this Appendix we give proofs of two statements used when deriving the hydrodynamic constitutive relation in Sec. 4.1.

The first statement is that the derivatives in the three building blocks $\nabla_{\perp\mu_1}\ldots\nabla_{\perp\mu_n}\log(s)$, $\nabla_{\perp\mu_1}\ldots\nabla_{\perp\mu_n}u_\nu$, and $\nabla_{\mu_1}\ldots\nabla_{\mu_{n-2}}\mathcal{R}$ can be commuted at the expense of introducing only

non-linear terms. To prove this, let's first recall that the commutator of covariant derivatives acts on a tensor as

$$[\nabla_\rho, \nabla_\sigma] t^{\mu_1...\mu_k}{}_{\nu_1...\nu_l} = R^{\mu_1}{}_{\lambda\rho\sigma} t^{\lambda\mu_2...\mu_k}{}_{\nu_1...\nu_l} + R^{\mu_2}{}_{\lambda\rho\sigma} t^{\mu_1\lambda...\mu_k}{}_{\nu_1...\nu_l} + \ldots + R^{\mu_k}{}_{\lambda\rho\sigma} t^{\mu_1...\lambda}{}_{\nu_1...\nu_l}$$
$$- R^\lambda{}_{\nu_1\rho\sigma} t^{\mu_1...\mu_k}{}_{\lambda\nu_2...\nu_l} - R^\lambda{}_{\nu_2\rho\sigma} t^{\mu_1...\mu_k}{}_{\nu_1\lambda...\nu_l} - \ldots - R^\lambda{}_{\nu_l\rho\sigma} t^{\mu_1...\mu_k}{}_{\nu_1...\lambda}, \tag{C.1}$$

where $R_{\mu\nu\rho\sigma}$ is the Riemann tensor (see e.g. [124]). As the Riemann tensor is linear in amplitudes, it is immediately clear that the covariant derivatives in $\nabla_{\mu_1} \ldots \nabla_{\mu_{n-2}} \mathcal{R}$ can be commuted at the expense of introducing non-linear terms. For the other two building blocks, we first use $\nabla_{\perp\mu} = \Delta_{\mu\nu} \nabla^\nu$ to obtain the following expression for the commutator of transverse derivatives acting on a tensor

$$[\nabla_{\perp\mu}, \nabla_{\perp\nu}] t^{\mu_1...\mu_k}{}_{\nu_1...\nu_l} = \Delta_{\mu\rho} \Delta_{\nu\sigma} [\nabla^\rho, \nabla^\sigma] t^{\mu_1...\mu_k}{}_{\nu_1...\nu_l} + \Delta_{\mu\rho} (\nabla^\rho \Delta_{\nu\sigma}) \left( \nabla^\sigma t^{\mu_1...\mu_k}{}_{\nu_1...\nu_l} \right)$$
$$- \Delta_{\nu\sigma} (\nabla^\sigma \Delta_{\mu\rho}) \left( \nabla^\rho t^{\mu_1...\mu_k}{}_{\nu_1...\nu_l} \right). \tag{C.2}$$

When the transverse derivatives act on a tensor which is linear in the amplitude, it is clear that this commutator is non-linear. The exceptional cases in the building blocks above are the rightmost transverse derivatives. These act directly on the hydrodynamic variables which are non-zero even in equilibrium. For these two special cases, Eq. (C.2) can be written

$$[\nabla_{\perp\mu}, \nabla_{\perp\nu}] t^{\mu_1...\mu_k}{}_{\nu_1...\nu_l} = \Delta_{\mu\rho} \Delta_{\nu\sigma} [\nabla^\rho, \nabla^\sigma] t^{\mu_1...\mu_k}{}_{\nu_1...\nu_l} + \text{non-linear}. \tag{C.3}$$

When the tensor is $\log(s)$, the first term on the right hand side vanishes identically as $\log(s)$ is a scalar. When the tensor is the fluid velocity, an explicit computation gives

$$\Delta_{\mu\rho} \Delta_{\nu\sigma} [\nabla^\rho, \nabla^\sigma] u_\lambda = \Delta_{\mu\rho} \Delta_{\nu\sigma} g_{\gamma\alpha} R^{\alpha\lambda\rho\sigma} u_\lambda = \frac{1}{2} \mathcal{R} (\Delta_{\mu\gamma} u^\sigma \Delta_{\nu\sigma} - \Delta_{\nu\gamma} u^\sigma \Delta_{\mu\sigma}) = 0, \tag{C.4}$$

where we used the expression

$$R_{\mu\nu\rho\sigma} = \frac{1}{2} \mathcal{R} (g_{\mu\rho} g_{\nu\sigma} - g_{\mu\sigma} g_{\nu\rho}), \tag{C.5}$$

for the Riemann tensor in (1+1) dimensions. Therefore even in these special cases the transverse derivatives can be commuted at the cost of introducing only non-linear terms.

The second statement is that the local conservation equations (4.7) imply that

$$\nabla_\perp^n \log(s) = -c_s^{-2} \nabla_\perp^{n-2} D (\nabla_\perp \cdot u) + c_s^{-2} \nabla_\perp^{n-2} \mathcal{R} + \ldots, \tag{C.6}$$

where the ... denote higher-derivative or non-linear terms. To prove this, we first act with $\nabla_{\perp\mu}$ on the second equation in (4.7) to obtain

$$\nabla_\perp^2 \log(s) = -c_s^{-2} \nabla_\perp^\mu D u_\mu + \text{higher-derivative terms}. \tag{C.7}$$

The commutator of the derivatives acting on the right hand side is

$$
\begin{aligned}
\left[\nabla_\perp^\mu, D\right] u_\mu &= \Delta^{\mu\nu} u^\rho \left[\nabla_\nu, \nabla_\rho\right] u_\mu + \Delta^{\mu\nu} \left(\nabla_\nu u^\rho\right)\left(\nabla_\rho u_\mu\right) - u^\rho \left(\nabla_\rho \Delta^{\mu\nu}\right)\left(\nabla_\nu u_\mu\right) \\
&= \Delta^{\mu\nu} u^\rho R^\sigma{}_{\mu\rho\nu} u_\sigma + \text{non-linear} \\
&= -\mathcal{R} + \text{non-linear},
\end{aligned}
\tag{C.8}
$$

where on the second line we used the commutator (C.1) of covariant derivatives and on the third line we used the expression (C.5) for the Riemann tensor in (1+1) dimensions. Commuting the derivatives in (C.7) using (C.8) and then acting on both sides with $\nabla_\perp^{n-2}$ gives the result (C.6).

# D   Viscosity of holographic theories

In this Appendix we will consider some explicit examples of theories where we expect the large$-c$ hydrodynamics described in the main text to be valid. These are holographic theories of three-dimensional gravity coupled to matter. For these theories we will compute the viscosity explicitly from first principles and show that it agrees with the result (4.22) argued for in Sec. 4. See [45, 125, 126] for textbook introductions to holographic theories and simple examples of the computation of their hydrodynamic transport coefficients.

## D.1   The equilibrium state

We consider three dimensional theories of gravity with action

$$
S = \frac{1}{16\pi G} \int d^3 x \sqrt{-g} \left(\mathcal{R} - \frac{1}{2}\partial_\mu \phi \partial^\mu \phi + V(\phi)\right) + S_{\text{bdy}},
\tag{D.1}
$$

where $G$ is Newton's constant, $\mathcal{R}$ is the Ricci scalar of the Lorentzian metric $g_{\mu\nu}$ and $\phi$ is a scalar field with potential

$$
V(\phi \to 0) = \frac{2}{L^2} - \frac{1}{2L^2}\Delta(\Delta - 2)\phi^2 + O(\phi^4),
\tag{D.2}
$$

and $L$ is a constant. $S_{\text{bdy}}$ is a boundary term needed to make the variational problem well-defined and the on-shell action finite: see [127] for details of this.

We study planar black hole solutions of this theory that are dual to the thermal state of a CFT deformed by a relevant scalar operator of dimension $1 < \Delta < 2$. We parameterize these solutions as

$$
ds^2 = -D(r)dt^2 + C(r)dx^2 + B(r)dr^2, \qquad\qquad \phi = \Phi(r),
\tag{D.3}
$$

and the equations of motion of the action (D.1) then require that

$$\frac{d}{dr} \log\left(\frac{C'}{\sqrt{BCD}}\right) = -\frac{C\Phi'^2}{C'}, \qquad \frac{d}{dr}\left(\frac{C^{3/2}(D/C)'}{\sqrt{BD}}\right) = 0,$$

$$\frac{d}{dr}\left(\sqrt{\frac{CD}{B}}\Phi'\right) = -\sqrt{BCD}\left.\frac{\partial V}{\partial \phi}\right|_{\phi=\Phi}, \tag{D.4}$$

where primes denote derivatives with respect to $r$. We furthermore impose the boundary conditions that there is an asymptotically AdS$_3$ boundary at $r = 0$ with

$$B(r \to 0) \to \frac{L^2}{r^2} + \ldots, \qquad C(r \to 0) \to \frac{L^2}{r^2} + \ldots,$$

$$D(r \to 0) \to \frac{L^2}{r^2} + \ldots, \qquad \Phi(r \to 0) \to \frac{\sqrt{12}\pi}{1-\Delta}\lambda r^{2-\Delta} + \ldots, \tag{D.5}$$

and a horizon at $r = r_0$ with

$$B(r \to r_0) \to \frac{b}{4\pi T(r_0 - r)} + \ldots, \qquad C(r \to r_0) \to (4Gs)^2 + \ldots,$$

$$D(r \to r_0) \to 4\pi Tb(r_0 - r) + \ldots, \qquad \Phi(r \to r_0) \to \Phi_0 + \ldots, \tag{D.6}$$

where $\lambda$, $T$, $s$, $b$ and $\Phi_0$ are constants. $s$ and $T$ are the entropy density and temperature of the state, and $\lambda$ is the relevant coupling that deforms the CFT.[15] After specifying $V(\phi)$ exactly, solving this set of equations yields the thermodynamic relations such as $s(T, \lambda)$ that characterize the equilibrium state.

For the special case $\Phi = 0$ there is the BTZ black hole solution

$$B(r) = \frac{L^2}{r^2 f(r)}, \qquad C(r) = \frac{L^2}{r^2}, \qquad D(r) = \frac{L^2}{r^2}f(r), \qquad f(r) = 1 - \frac{r^2}{r_0^2}. \tag{D.7}$$

By examining the boundary condition (D.5) for $\Phi$, we see that this corresponds to the case $\lambda = 0$ (an undeformed CFT). This black hole has temperature $T = 1/(2\pi r_0)$ and $s/T = \pi L/(2G)$. Comparing this to the result for the entropy density of an undeformed CFT in Sec. 2 yields the classic holographic expression for the central charge $c = 3L/(2G)$.

## D.2   Viscosity formula and probe limit

To compute the viscosity we need to determine two-point functions of the stress tensor in the thermal states we have just described. Although fundamentally this requires the study of black hole perturbations, the final result can be expressed in terms of the equilibrium state as [38]

$$\frac{\zeta}{s} = \frac{s^2}{4\pi}\left(\frac{\partial \Phi_0}{\partial s}\right)_\lambda^2. \tag{D.8}$$

---

[15]The prefactor in (D.5) corresponds to turning on a source $\lambda$ for an operator with vacuum correlator $c\,|x - x'|^{-2\Delta}$, consistent with the normalizations in the main text.

This formula is exact. In [38] it is presented for holographic theories of general dimensionality, with $s$ on the left hand side replaced by $4\pi\eta$ with $\eta$ the shear viscosity. There is a subtlety in (1+1) dimensions where there is no shear viscosity. However, we have checked by an independent method (analogous to that in [128], see also [129]) that the formula (D.8) does indeed hold for field theories in (1+1) dimensions.

Substituting the BTZ solution ($\Phi = 0$) into the formula (D.8) trivially produces the correct viscosity $\zeta = 0$ of an undeformed CFT. When conformal symmetry is broken, we no longer have an exact expression for $\Phi_0$. But we can make progress when $\bar{\lambda}$ is small by assuming that this means $\Phi$ remains small enough everywhere outside the black hole that we can neglect its backreaction on the BTZ metric. Intuitively, the high temperature means that the region where scalar field corrections become large is hidden behind the horizon. This is the approximation made in [37] and [38] which, written in our conventions, reproduces the result (4.22) we have argued for in the main text.

It is instructive to see explicitly how this works. Treating $\Phi(r) = \bar{\lambda}\delta\Phi(r) + \ldots$ as a small perturbation on the fixed BTZ background allows us to linearize the third equation of motion in (D.4) to give

$$\frac{d}{dr}\left(\frac{f(r)}{r}\delta\Phi'(r)\right) - \frac{\Delta(\Delta - 2)}{r^3}\delta\Phi(r) = 0. \tag{D.9}$$

The solution of this equation that obeys the boundary conditions in the previous Section is

$$\delta\Phi(r) = \frac{\sqrt{12}\pi}{(1 - \Delta)}\frac{(2\pi)^{\Delta-2}\Gamma(\frac{\Delta}{2})^2}{\Gamma(\Delta - 1)} \,_2F_1\left(1 - \frac{\Delta}{2}, \frac{\Delta}{2}, 1; 1 - \frac{r_0^2}{r^2}\right), \tag{D.10}$$

and thus

$$\Phi_0 = \frac{\sqrt{12}\pi}{(1 - \Delta)}\left(\frac{6s}{c}\right)^{\Delta-2}\frac{\Gamma(\frac{\Delta}{2})^2}{\Gamma(\Delta - 1)}\lambda + \ldots, \tag{D.11}$$

at small $\lambda$. Substituting this into the expression (D.8) gives the expression (4.22) for the viscosity proposed in the main text.

However, on its own this calculation is not a proof of the result (4.22) for these theories. The linearized solution (D.10) captures exactly the conformal expression for the thermal two-point function of the deforming operator. And so in assuming that the small $\bar{\lambda}$ viscosity is given by this solution, we are really making the same assumption as in the main text.

## D.3   Exact results for viscosity

To truly verify the result (4.22) for holographic theories, we will now go beyond the linearized solution and solve the full non-linear equations of motion (D.4). This can only be done numerically.

For definiteness, and following [130], we considered the family of potentials

$$V(\phi) = \frac{1}{2}\left(W^2 - \left(\frac{\partial W}{\partial \phi}\right)^2\right), \qquad W(\phi) = -2 + \frac{\Delta - 2}{2}\phi^2 + \alpha\phi^4. \tag{D.12}$$

To solve the equations of motion numerically we used the procedure described in Sec. 3 of [130], which builds on [131]. For a given potential we obtained black hole solutions for different values of $s$ and $\lambda$, and then computed the viscosity by numerically evaluating the right hand side of the exact formula (D.8). The speed of sound for each solution was obtained numerically as described in [130].

In Fig. 6 we show the results we obtained for the ratio $\zeta/(s(1 - c_s^2)$ for four different potentials. In all cases the small $\bar{\lambda}$ behaviour agrees with the expression (4.23) proposed in the main text. The value of $\bar{\lambda}$ where corrections to this expression become important – and the effect they have – is sensitive to the details of the potential. Thermodynamic properties of the equilibrium state for some of these potentials can be found in [130].

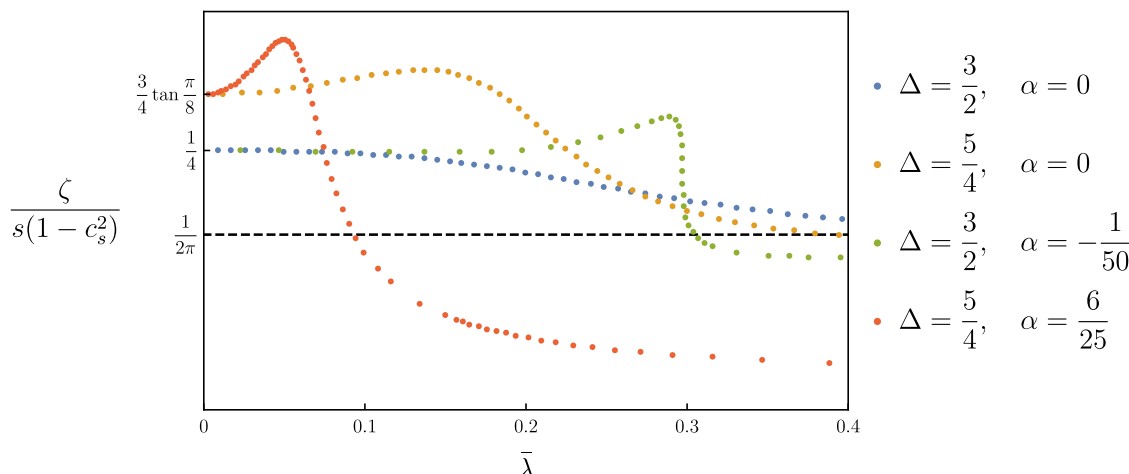

**Figure 6:** Numerical results for the the ratio $\zeta/(s(1 - c_s^2)$ as a function of $\bar{\lambda}$ for four different potentials. All cases show agreement with Eq. (4.23) at small $\bar{\lambda}$.

The analysis above can easily be extended to the case of deforming UV CFTs with an operator of dimension $0 < \Delta < 1$ by considering an alternate quantization of the scalar field $\phi$. Holographic examples of IR CFTs deformed by operators with arbitrary $\Delta > 2$ can also be generated by choosing the potential appropriately – it would be interesting to repeat our analysis for these cases.

# E    Causality of large-$c$ hydrodynamics

In this Appendix we will examine the resummed hydrodynamic dispersion relations and show that tensions with causality arise only at the wavenumbers where hydrodynamics starts to break down. This is a check that the large-$c$ theory of hydrodynamics we have proposed makes sense within its regime of validity.

In relativistic QFTs, microcausality – the fact that space-like separated operators commute – requires that retarded Green's functions are analytic in the region where $\operatorname{Im} p^\mu$ is a time-like vector [132]. Any non-analyticity, such as poles, must therefore satisfy

$$\operatorname{Im}\left(\omega_\pm(k)\right) \leq \left|\operatorname{Im}\left(k\right)\right|. \tag{E.1}$$

In [46, 133] the inequality (E.1) was used to derive causal bounds on the values of individual transport coefficients. However, as we have an expression for the full dispersion relation at small $\bar\lambda$ we will work directly with the fundamental inequality (E.1). We will discuss only the right-moving mode $\omega_+(k)$ as the conditions arising from $\omega_-(k)$ are identical.

The condition (E.1) for the right-moving mode $\omega_+(k)$ is in fact two different inequalities, one for each sign of $\operatorname{Im}(k)$. It will be instructive to consider first the case of purely imaginary $k = i\kappa$, $\kappa \in \mathbb{R}$. For negative $\kappa$, the causality constraint on the correction to the thermal CFT dispersion relation is

$$\operatorname{Re}\Gamma_+(i\kappa) \geq -2, \qquad \kappa \leq 0. \tag{E.2}$$

For this to be violated, the 'small' correction $\Gamma_+$ must be parametrically large. We saw in the main text that $\Gamma_+$ has a pole at $\kappa = -\pi\Delta T$, which sets the radius of convergence of the dispersion relation. This same pole produces a parametrically large $\Gamma_+$ at $\kappa = -\pi\Delta T + O(\bar\lambda^2)$ and so the causality inequality (E.2) starts to be violated precisely when hydrodynamics breaks down.

For our hydrodynamic theory to be self-consistent, we also require the corresponding causality constraint for $\kappa \geq 0$ to be satisfied everywhere within the radius of convergence. Specifically this requires that

$$\operatorname{Re}\frac{\Gamma_+(i\kappa)}{\bar\lambda^2} = \frac{(2-\Delta)^2}{2(\Delta-1)} \frac{\alpha_\Delta}{1 - \left(\frac{\kappa}{2\pi T}\right)^2} \left( \frac{\Gamma\left(\frac{\Delta}{2} + \frac{\kappa}{2\pi T}\right) \Gamma\left(\frac{\Delta}{2} - \frac{\kappa}{2\pi T}\right)}{\Gamma\left(\frac{\Delta}{2}\right)^2} \frac{\sin\left(\frac{\pi\Delta}{2} - \frac{\kappa}{2T}\right)}{\sin\left(\frac{\pi\Delta}{2}\right)} - \frac{\Delta}{2-\Delta} \right) \leq 0,$$
$$\tag{E.3}$$

for $0 \leq \kappa \lesssim \pi\Delta T$. This quantity is independent of $\bar\lambda$ and, by plotting it, it is straightforward to verify that (E.3) is indeed satisfied for all $0 < \Delta < 3$.

Considering the general case of complex $k$ does not affect these conclusions: the first tensions with the causality inequality (E.1) arise for $k \approx -i\pi\Delta T$ where hydrodynamics starts to break down.

We close by noting a surprising feature of our hydrodynamic theory. We can define the phase velocity and group velocity of the hydrodynamic sound wave as

$$v_{\text{phase}}(k) = \text{Re}\left(\frac{\omega_+(k)}{k}\right) = 1 - \bar{\lambda}^2 \delta v_{\text{phase}}(k) + \ldots,$$
$$v_{\text{group}}(k) = \text{Re}\left(\frac{d\omega_+(k)}{dk}\right) = 1 - \bar{\lambda}^2 \delta v_{\text{group}}(k) + \ldots,$$
(E.4)

where $k$ is real. From our dispersion relation (4.29), it is straightforward to extract the leading deviations of these quantities from the speed of light at small $\bar{\lambda}^2$ and these are shown in Fig. 7. For all $0 < \Delta < 3$, the phase velocity is subluminal. However, for $0 < \Delta < 2$ the

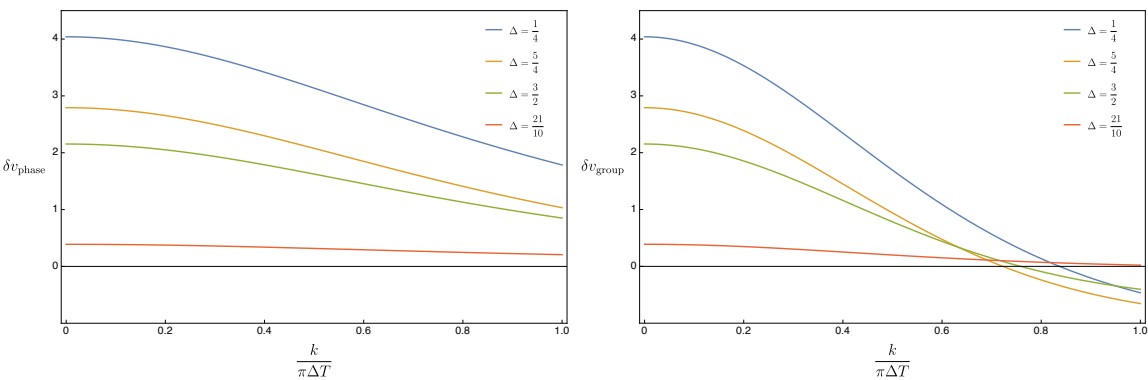

**Figure 7:** Plots of the leading corrections to the phase and group velocities of the hydrodynamic sound mode, extracted from Eq. (4.29).

group velocity of the sound wave becomes superluminal. Although this happens at large wavenumbers, these are still within the range of applicability of hydrodynamics. This is despite the fact the theory satisfies the fundamental causality requirement (E.1). It would be very interesting to understand better this surprising feature.

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
