# Peer review of "Universal thermalization dynamics in (1+1)d QFTs"

_SciPost Physics_

## Round 1 · Referee Report · Anonymous (Referee 1) · 2024-12-18

Strengths

Important and timely subject
Calculations are novel and support very well the main results
One of the main results is surprising and seems important
The scaling analysis in Section 3.3 is quite nice!

Weaknesses

The derivations are missing equations, so some are not so clear
One of the main result is in a sense quite expected

Report

In this paper the authors consider the perturbation theory for QFT at high and low temperatures, assuming that the QFT both in the UV and IR points is described by a nontrivial CFT. Analysing the conformal perturbation theory for real-time correlation functions, the attempt to establish the time scale at which hydrodynamics is supposed to arise. The idea is that the hydrodynamics of the critical points is significantly different from that of the QFT, thus the time at which conformal perturbation theory breaks should indicate the time at which hydrodynamics emerges.

The time scale found is proportional to $T^a/\lambda^2$ where $\lambda$ is the coupling constant, where $a$ is a power obtained from the fixed point theory by dimensional analysis.

The authors also provide an in-depth analysis in the case of large-$c$ theories, where perturbation theory simplifies slightly. In particular, they provide a universal generating functions for all-order transport coefficients in this case, and use the breaking of hydrodynamics (instead of perturbation theory) to establish the time scale, getting agreement with the perturbative analysis.

There are quite a few ideas and results in this paper. Let me discuss my opinion of the main results.

The first point is that a time scale of $1/\lambda^2$ is I believe quite expected. Indeed this is the time scale at which “Boltzmann-equation” dynamics occurs, in a very general way. For instance, for any integrable system, the breaking of the system by any small perturbation $\lambda$ leads to a slow dynamics on time scales proportional to $1/\lambda^2$. See for instance [Phys. Rev. X 9, 021027 (2019); Phys. Rev. Lett. 127, 130601 (2021)]. These are extensions to arbitrary “non-perturbed” model (including strongly-coupled non-perturbed models) of the usual “kinetic arguments”. It seems very plausible that such general arguments can be applied to QFT seen as CFT perturbations. Thus, the statement at the beginning of the introduction, about the question of how long a system takes to thermalise, is not entirely representative of the current understanding. I emphasise also that the result is general, and does not require the “non-perturbed theory” to be weakly coupled; again in contradiction to what is stated in the introduction. My impression is that the authors wanted to argue that the usual arguments (kinetic theory arguments) required weak coupling for the unperturbed theory, while they could do CFT with strong coupling; however in the modern understanding of “kinetic theory” weak coupling is not required. At least, this is my understanding.

This being said, I think it is very interesting to obtain this time scale from the breaking of conformal perturbation theory. The typical ways involve leading-order perturbation theory and assumptions about the slow variation of the state. Here, instead, the two-point function is directly analysed (this had been done before, but not for conformal perturbation theory, which indeed then admits strong-coupling non-perturbed models). It is only a conjecture, that the breakdown time of perturbation theory corresponds to the time of emergence of hydrodynamics. But it is plausible. Further, the argument about the breakdown of the exact (conjectured) all-order hydrodynamic form of the trace two-point function in the $c=\infty$ region supports the obtained time scale. So, although I find the main result not surprising, I think the derivation provides new information and new ideas.

The second point is that the results concerning exact transport coefficients in the infinite-$c$ region seems to be very interesting. It is a conjecture, as the authors emphasise, but I believe the exchange of limits involved in this conjecture is indeed believable. So, this other main result is perhaps less “expected” and even more interesting, although it is “only” for the infinite-$c$ region.

The paper is written in a way that is mostly readable, however I find some of the derivations somewhat hard to follow and imprecise (see below); more equations would be useful.

Overall, thus, I believe this is a strong paper. The arguments for why the main result is interesting are a bit weak, and miss some known results in the literature, which make one of the main results less surprising. Nevertheless, the calculations and derivations are very interesting, and the other main results seems to be very strong. Thus, I would say the paper, with discussion adjusted to clarify the context, will meet the criteria for Scipost Physics.

Things to change:

Introduction: Be more accurate concerning current understanding of how long to thermalise, etc, including the fact that $1/\lambda^2$ is expected and that weak coupling is not necessary, and refer to papers mentioned (and / or others if found to be relevant).

Introduction: Make more clear that the QFT is assumed to be described by a nontrivial CFT at both UV and IR. This is important, because at low temperature, if there is a gap (i.e. the CFT is the trivial one-dimensional CFT spanned by the operator identity), as is the case in many one-dimensional QFT’s studied in the literature, the results are very different.

Page 7: “We will mostly focus on the generic case given in Eq. (2.7).” Mostly? Not always? Could you be more specific?

Page 8: $G^E$ or $G_E$? The former is a typo?

Page 8: why not define $O_{\rm eq}$ mathematically? It is just put in text and explained in words, which it very inefficient.

Page 8: eq 2.11 seems to be important. However the derivation presented on pages 7-8 leading to it is very hard to follow. It is said in words, with some equations in App A. However, it is not clear where 2.11 actually come from. At least the full derivation of 2.11 should be given in App A and referred to in the text.

Section 3.2: I can more or less follow the arguments made in section 3.2, which are very interesting, but it would be useful to put more key equations to support the arguments. Currently, I could not, with the limited time I had, follow all derivations because I would have had to write the equations.

Pages 14-15, it is not clear what “schematic” means in equation 3.17 and 3.18. Does it mean that coefficients are not all 1? Please be more precise there. Also, it is unclear where these equations come from.

Page 15: the terminology double-twist and higher-twist is not so universal. Please explain.

Pages 14-15: in fact I find the paragraph “large-c scaling” completely unclear and hard to follow. Perhaps it is better placed in section 4, and with more explanations?

Section 4: I could not follow as well, however all arguments, including the exchange of limits necessary for the conjecture, seem plausible.

Positive note: The scaling analysis in Section 3.3 is quite nice!

Requested changes

(repeated from report)

Introduction: Be more accurate concerning current understanding of how long to thermalise, etc, including the fact that $1/\lambda^2$ is expected and that weak coupling is not necessary, and refer to papers mentioned (and / or others if found to be relevant).

Introduction: Make more clear that the QFT is assumed to be described by a nontrivial CFT at both UV and IR. This is important, because at low temperature, if there is a gap (i.e. the CFT is the trivial one-dimensional CFT spanned by the operator identity), as is the case in many one-dimensional QFT’s studied in the literature, the results are very different.

Page 7: “We will mostly focus on the generic case given in Eq. (2.7).” Mostly? Not always? Could you be more specific?

Page 8: $G^E$ or $G_E$? The former is a typo?

Page 8: why not define $O_{\rm eq}$ mathematically? It is just put in text and explained in words, which it very inefficient.

Page 8: eq 2.11 seems to be important. However the derivation presented on pages 7-8 leading to it is very hard to follow. It is said in words, with some equations in App A. However, it is not clear where 2.11 actually come from. At least the full derivation of 2.11 should be given in App A and referred to in the text.

Section 3.2: I can more or less follow the arguments made in section 3.2, which are very interesting, but it would be useful to put more key equations to support the arguments. Currently, I could not, with the limited time I had, follow all derivations because I would have had to write the equations.

Pages 14-15, it is not clear what “schematic” means in equation 3.17 and 3.18. Does it mean that coefficients are not all 1? Please be more precise there. Also, it is unclear where these equations come from.

Page 15: the terminology double-twist and higher-twist is not so universal. Please explain.

Pages 14-15: in fact I find the paragraph “large-c scaling” completely unclear and hard to follow. Perhaps it is better placed in section 4, and with more explanations?

Section 4: I could not follow as well, however all arguments, including the exchange of limits necessary for the conjecture, seem plausible.

Recommendation

Publish (meets expectations and criteria for this Journal)

---

## Round 1 · Referee Report · Anonymous (Referee 2) · 2024-12-19

Strengths

1) Identifies a mechanism for emergence of hydrodynamics from a CFT fixed point in 2d 2) Proposes an all-order resummation of the hydrodynamic expansion in the large-c limit

Weaknesses

1) The results are shown to be valid only under certain assumptions

Report

The paper examines the emergence of hydrodynamics in a 1+1-dimensional QFT that is obtained by a small relevant (irrelevant) perturbation of an UV (IR) CFT by a scalar operator. The authors show that the hydrodynamic behaviour can only appear as a consequence of conformal perturbation theory becoming unreliable at long times, and identify the channel that gives the leading contribution to the stress-energy tensor correlators away from the light cone and close to the sound-front propagation. In this way they identify a universal time-scale that separates the early-time CFT dynamics from the late-time hydrodynamics.
In addition, the authors conjecture an all-order resummation of the hydrodynamic gradient expansion, for the class of theories described above and when the CFT has a large central charge, and find that the radius of convergence of the resummed expansion defines the same time-scale as was found in the previous section, now reached from the late-time expansion instead of the perturbative early-time one.

The paper is well-written and discusses carefully the assumptions and caveats behind their general statements. It is certainly a worthy contribution to the large body of work concerning the mechanism of thermalization in interacting field theories. I recommend its publication.

Requested changes

1) In the introduction, the authors seem to identify the concepts of thermalization and hydrodynamics. As they note much later, in the discussion section, a CFT can still thermalize (e.g. in the sense of correlators approaching the thermal value), even though it does not admit hydrodynamics. It would be better to be more clear about this already in the introduction.

2) The authors invoke frequently the notion of breakdown of perturbation theory. Also here I think that they should clarify in what sense they mean this: to my understanding, it is simply that in the hydrodynamic regime the effective coupling becomes large, but the resummation to all-orders of the CPT can still describe the physics without the need of non-perturbative effects.

3) When one considers higher orders in CPT, generically one has to consider that the RG flow will introduce mixing of different operators; is there an argument for why it is sufficient to consider the contribution of a single operator as illustrated in Fig. 1? Relatedly, in the large-c limit illustrated in Fig. 2, can one ignore diagrams with O O fusing into O’, and O O’ fusing into T?

4) On page 18, the authors advocate for a definition of the equilibration time purely within hydrodynamics, which would be more universal than the one defined from perturbation theory. However with this definition, it would still depend in principle on the full set of transport coefficients, which in turn contain the information of the UV theory, so it is not clear to me to what extent it would be more universal.

5) On page 25 they write “we do not expect the TTbar deformation to generate viscous effects”. In fact ref [24] found a non-vanishing momentum diffusion in TTbar-deformed CFT at leading order in CPT, see eq (15)-(16).

6) In appendix B, they state that the UV divergence in the Fourier transform of the scalar two-point function for $\Delta \geq1$ can be reabsorbed in a local counterterm. How does it work, since the counterterm action they write would generate a contact term in the correlator, while the divergence comes from the light-cone ?

7) In eq. (4.20), the scalar correlator on the right-hand side should determine all the transport coefficients, but those are encoded in two functions of $\omega,k$: can the author explain in more detail how one single equation fixes both functions?

Recommendation

Publish (easily meets expectations and criteria for this Journal; among top 50%)

---

## Editorial Decision

resubmitted